# Measuring global human accessibility to essential daily necessities and services

Shengbiao Wu [1], Bin Chen [1,2,3] ✉, Jiafu An [2,3,4], Andrew Nelson [5], Fan Dai[3], Chen Lin [2,3,6] & Peng Gong [2,3,7] ✉

Equitable access to daily necessities and services is crucial for enhancing human quality of life and achieving the United Nations' Sustainable Development Goals. However, knowledge about global access to these essential resources remains limited and fragmented, due to the absence of a comprehensive infrastructure inventory and scalable accessibility measures. Here we compile a global database of points of interest to represent six essential infrastructure categories—living, healthcare, education, entertainment, public transit, and working. We use refined 30-meter resolution friction surface data to map travel time to these critical infrastructures as a proxy for accessibility across the urban-rural continuum and assess disparities across geographic, urbanization, and socio-economic contexts. Our results reveal uneven access in infrastructure availability, per capita distribution, and travel time. Globally, 62.8% (3.08 billion) and 82.5 % (4.04 billion) of urban residents live within a 15- and 30-minute walk of essential resources, respectively. These results highlight the need to optimize strategies for planning, allocation, and management of critical infrastructure to promote inclusive and sustainable development.

Human daily activities depend on the availability and allocation of essential amenities, which provide resources, services, and opportunities for work, communication, sustenance, relaxation, and economic wealth generation, as well as promoting health and well-being[1,2]. Equitable access to those daily necessities and services (i.e., individuals have equal opportunity to reach amenities, regardless of socio-economic disparities in need) is crucial for enhancing life satisfaction, happiness, and community prosperity[3], aligning with the United Nations' Sustainable Development Goals (SDGs)[4]. Notably, 72% of the SDGs' targets are directly or indirectly linked to infrastructure[5]. However, access to essential infrastructure is often compromised by social, economic, and political factors across different countries and regions[6,7]. Many theoretical and empirical studies have shown that dominant individuals and entities, such as high-income individuals and

countries[8], have disproportionately good access to resources and goods, including education, healthcare, and food, compared to their more vulnerable counterparts[9,10]. Despite widespread recognition of disparities in access to specific amenities such as healthcare facilities[11], a comprehensive understanding of global human access to essential daily necessities and services remains limited and particularly fragmented due to the lack of a complete inventory of these essential infrastructures and a scalable measurement of accessibility.

Almost 70% of the world's population is concentrated in urban areas covering only 1–2% of the Earth's land surface[12]. These areas serve as socioeconomic, cultural, and technological hubs and are expected to enhance residents' quality of life by providing diverse service amenities and convenient transport networks[13–15]. To achieve this objective, urban planners and policymakers have embraced the "*x*-

[1]Future Urbanity & Sustainable Environment (FUSE) Lab, Division of Landscape Architecture, Department of Architecture, Faculty of Architecture, The University of Hong Kong, Hong Kong SAR, China. [2]Urban Systems Institute, The University of Hong Kong, Hong Kong SAR, China. [3]Institute for Climate and Carbon Neutrality, The University of Hong Kong, Hong Kong SAR, China. [4]Department of Real Estate and Construction, Faculty of Architecture, The University of Hong Kong, Hong Kong SAR, China. [5]Department of Natural Resources, Faculty of Geo-Information Science and Earth Observation (ITC) of the University of Twente, Enschede, The Netherlands. [6]Faculty of Business and Economics, The University of Hong Kong, Hong Kong SAR, China. [7]Department of Geography and Department of Earth and Planetary Sciences, The University of Hong Kong, Hong Kong SAR, China. ✉e-mail: binley.chen@hku.hk; penggong@hku.hk

minute city" concept, advocating that residents should be able to access daily essentials within a short walk, bike ride or other mode of public transit within x min. This reflects the chrono-urbanism theory that urban life quality inversely correlates with commuting time[16,17]. In particular, the "15-minute city" model has become popular and is increasingly adopted by cities worldwide to promote active transportation, reduce greenhouse gas emissions, and foster inclusive, cohesive, affordable, and sustainable urban environments[18–21]. The 2024 Global Sustainable Development Report[22] involves the "urban population with access to points of interest (POIs) within a 15-min walk" as a proxy for specific targets of the 11th SDG. However, this proxy map currently covers only 38 countries with qualitative description whether a country has achieved this objective or not, leaving significant gaps in our comprehension of the pattern and extent of global human access to daily necessities and services[23]. A recent study assessed and visualized accessibility scores based on the 15-min city model for cities worldwide[24]. However, this analysis represents a singular urban scenario, but not to incorporate urban-rural gradients and the adaptable x-min concepts that better reflect diverse contextual variations.

A central focus of the "x-minute city" concept is developing holistic methods or metrics to evaluate accessibility to essential services, considering factors such as service availability, road networks, active mobility options, and socioeconomic diversity[24,25]. Location-based approaches—such as proximity-based measures and the two-step floating catchment area method—are widely used to assess accessibility, quantifying service availability within specific travel distances or time[26]. More advanced models, such as spatial-temporal interaction frameworks, incorporate additional dimensions like service capability, attractiveness, user preferences, and human mobility patterns to better reflect actual usage of amenities[17]. Among these metrics, proximity-based measures (e.g., accessibility or proximity) remain the most common due to their simplicity and efficiency. There are two primary approaches of characterizing the accessibility or proximity of essential amenities: the isochrone[18,25,26] and travel-time mapping[2,7,11]. The isochrone-based method maps walking or riding zones for each amenity using open-source routing software, such as Openrouteservice[27] and Google Application Programming Interfaces (APIs)[28], by combining road networks and specific pedestrian or ride travel speeds (e.g., 5 km/h for pedestrians). These zones are then overlapped to extract accessibility regions—areas where all types of amenities are reachable within the predefined x-min timeframe. Although efficient for local street- or block-level accessibility quantification, this amenity-centric method is challenging for city- or global-scale mapping due to the complexity of road networks and the millions of services and amenities globally. In contrast, the travel-time-based approach is more suitable for continental- and global-scale mapping. This method calculates and saves a global friction surface as a raster layer and employs a least-cost-path algorithm to determine pixel-level time costs (i.e., travel time) to the nearest daily necessities and services. However, this approach is predominantly used for inter-city rather than intra-city travel time mapping due to the lack of high-spatial-resolution friction surface datasets in the existing literature.

To address the key challenges identified, we first integrate a suite of geospatial data layers, including the 10-m-resolution WorldCover land cover[29], and 30-m-resolution Advanced Land Observing Satellite (ALOS) World digital surface model (DSM)[30], and global road vector dataset[31] to create 30-m-resolution global friction surface datasets for two baseline transport modes (non-motorized and motorized), following a widely used framework[2,7]. We then generate spatially explicit vector layers for six categories of essential infrastructures—living, healthcare, education, entertainment, public transit, and working—to characterize residents' daily living necessities and services, utilizing data from the Overture[31] (globe) and Amap[32] (China) POI databases (Supplementary Tables 1–3). By integrating these two key components with global high-resolution gridded population density map from

WorldPop[33], we achieve a high-resolution estimate of global human access to daily necessities and services. Specifically, we aim to answer the following three questions. (1) How much travel time is required for urban residents to access essential daily necessities and services? (2) What are the global patterns of the "x-minute city" model, and how far we are from achieving it? (3) To what extent does disparity in the travel time to access daily necessities and services vary across geographic, urbanization, and socio-economic spectrums?

## Results

### Varying accessibility of daily necessities and services

We compiled a comprehensive global POI dataset, consisting of ~66 million records. We organized these records into six major categories of essential infrastructures as individual vector layers to represent daily necessities and services for human daily activities, specifically, living ($N = 41,394,922$), healthcare ($N = 4,261,898$), education ($N = 4,644,982$), entertainment ($N = 6,308,387$), public transit ($N = 4,486,765$), and working ($N = 4,619,809$) (Supplementary Fig. 1). By overlapping these POI layers with population distribution, we assess whether these infrastructures, as fundamental living necessities and resources, are equitably and optimally distributed and accessible to residents across various geographic and socio-economic spectrums (Fig. 1 and Supplementary Fig. 2).

At the national scale, residents living in countries such as the United States, Canada, Australia, New Zealand, and most European nations enjoy the highest accessibility to daily necessities and services, with a POI number per thousand residents exceeding 24. This is followed closely by countries such as Brazil, Poland, Latvia, Lithuania, Slovakia, Bulgaria, Thailand, and Malaysia with a POI number per thousand residents ranging from 16 to 24. In contrast, countries like China, Romania, South Africa, Bolivia, Colombia, and Peru have relatively low infrastructure density, with the value between 8 and 16. Meanwhile, countries in Africa, the Middle East, and Russia experience the least access, with a POI number per thousand residents below 8 (Fig. 1b). A state-level map provides further details on these geographic disparities (Fig. 1c). At the continental scale, residents in Australia/Oceania are most likely to access their living necessities and services with the highest infrastructure density (5.6 per thousand), followed by North America (3.8 per thousand), Europe (3.5 per thousand), South America (2.3 per thousand), Asia (1.0 per thousand), and Africa (0.2 thousand) (Fig. 1d). We also observe a contrasting difference in infrastructure accessibility between the Global South and Global North. Global North countries have a POI number per thousand residents of 4.0, which is four times higher than that of Global South countries at 0.9. Moreover, economic development significantly reshapes the global distribution of living amenities, as demonstrated by the consistent ranking pattern between income level and infrastructure density (Fig. 1d).

By integrating the global friction surface database with the six vector layers of essential infrastructures, we have mapped the global 30-m-resolution population-weighted travel time across urban areas, as defined by the latest Global Urban Boundary (GUB) product[34]. The time calculated was further aggregated at state and country levels. Results reveal consistent geographic patterns in global travel time across two baseline transport modes: non-motorized (Fig. 2) and motorized transportation (Supplementary Fig. 3), as well as their combinations (Supplementary Fig. 4). In most countries and states, the average travel time to access the six types of essential daily necessities and services is under 15 min using non-motorized transport, indicating that residents in these areas can readily access living essentials through daily foot activity (Fig. 2a, b). Travel time, influenced by the interplay between transportation networks and infrastructure distribution, varies across geography, socio-economic status, and income levels (Fig. 2c). For example, residents in South America experience the shortest travel time of 10.0 min, followed by Europe (13.5 min), North

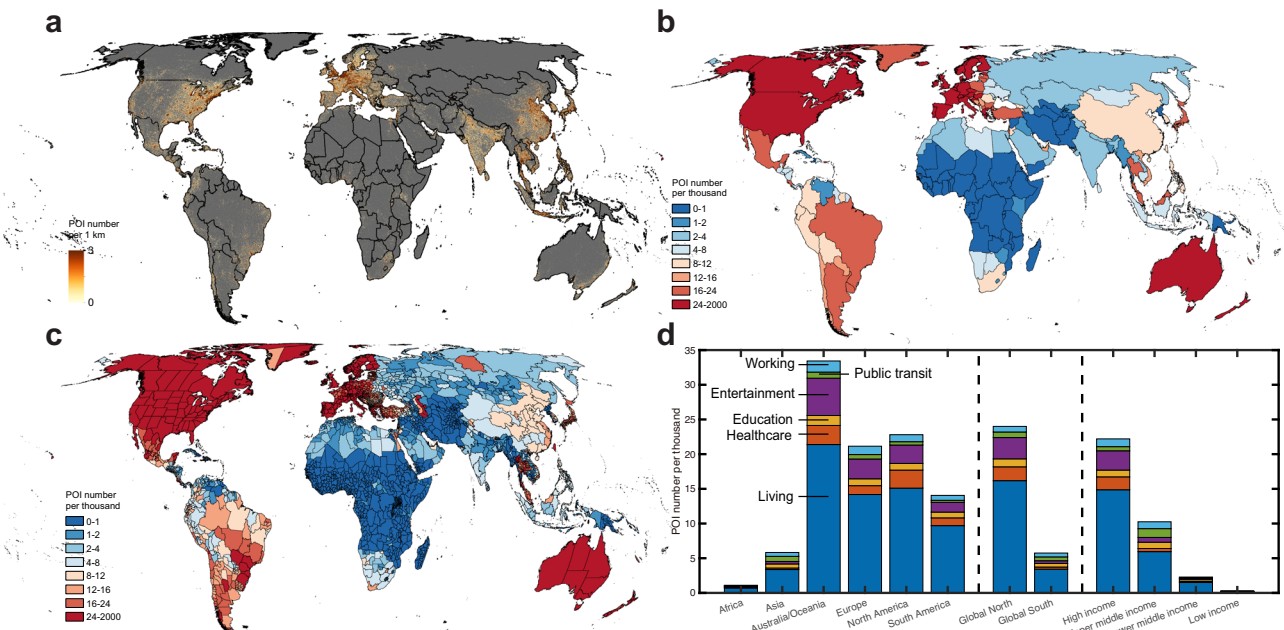

**Fig. 1 | Global distribution of point of interests (POIs) measuring six critical infrastructures in resident living services and amenities. a** 1-km-resolution density maps of total POI number of six critical infrastructures. Maps of total POI number per thousand residents across country (**b**) and state (**c**) levels. **d** Composition of different POI types per thousand residents across different continent (Africa, Asia, Australia/Oceania, Europe, North America, and South America), urbanization (Global North and Global South), and income (high, upper middle, lower middle, and low incomes) contexts. Income information is from the World Bank database according to 2022 gross national income (GNI) per capita: low income, 1135 USD or less; lower middle income, 1136–4465 USD; upper middle income, 4466–13,845 USD; and high income, 13,846 USD or more. Population density is extracted from the 100-m resolution WorldPop dataset. The boundaries vectors for country, state, continent, and Global North and Global South are from the Global Administrative Unit Layers (GAUL) dataset provided by the Food and Agriculture Organization (FAO) of the United Stations (https://data.apps.fao.org).

America (14.6 min), Australia/Oceania (16.2 min), Asia (28.0 min), and Africa (45.9 min). The Global North has an average travel time of 15.5 min, which is only half of the 30.8 min for the Global South. Additionally, the correlation analysis between travel time and Gross Domestic Product (GDP) per capita clearly illustrates the impact of economic factors (Fig. 2d). For example, Iran, with a GDP per capita of 4669 USD for 2022, experiences an average travel time of 55.4 min, which is three times longer than Monaco's 16.7 min, where the GDP per capita is 240,862 USD. These disparities in travel time are consistent across different quality levels of global POI inventories (Supplementary Figs. 5 and 6).

## Global pattern of the "x-minute city" model

By overlapping gridded population data with travel time mapping, we quantified the number and proportion of the urban population at the country level, adhering to the "x-minute city" model using four different time thresholds (10, 15, 20, and 30 min) for both non-motorized (Fig. 3) and motorized (Supplementary Fig. 7) transports, revealing consistent spatial patterns. Generally, both the number and proportion of the urban population increase with the time threshold, indicating that residents can expand their travel distances or areas to access and utilize essential daily necessities and services.

Under the widely used "15-minute city" model, only a few countries (e.g., Peru, United Kingdom, Netherlands, Greece, Japan, Thailand, and the Philippines) enable over 80% of their population to access daily necessities and services within a 15-min walk (Fig. 3b). Globally, a total of 3.08 billion urban residents (62.8% of global urban population) live within a 15-min reach of daily living essentials (Table 1), with 2.50 billion (81.2%) in the Global North and 0.58 billion (18.8%) in the Global South (Fig. 3c, d). This population is unevenly distributed across different income levels, with 0.74 billion (24.0%) in high-income (gross national income (GNI) per capita ≥13,846 USD, World Bank

2022), 1.26 billion (40.9%) in upper-middle-income (4466–13,845 USD), 0.99 billion (32.1%) in lower-middle-income (1136–4465 USD), and 0.09 billion (3.0%) in low-income (≤1135 USD) countries. When the time threshold is increased to 30 min, the number of countries where 80% of the urban population can access daily living necessities and services increases to 93 (or 33.7% of the 276 administrative units at the country and region level defined in the Global Administrative Unit Layers of the Food and Agriculture Organization of the United Nations in 2015[35]) (Fig. 3d). This totals 4.04 billion residents, with 3.33 billion (82.6%) living in the Global North and 0.71 billion (17.4%) in the Global South (Fig. 3c, d). Local examples across four cities (Chicago, Washington, Berlin, and Paris) show large heterogeneous in travel time maps that are shaped by spatial configurations of infrastructure amenities and transportation networks (Supplementary Figs. 8–15).

## Urban-rural travel time contrasts

We examined the difference in travel time between urban and rural areas, which represent the contrast in economic and environmental contexts and is significantly influenced by the geographical distribution of population density, transportation networks, and essential infrastructures (Fig. 4 and Supplementary Fig. 16). Our analysis reveals that among 276 administrative units defined at country and region level, 124 (44.9%) have travel time to daily necessities and services within 15 min, 44 (15.9%) between 15–30 min, 18 (6.5%) between 30–60 min, and 9 (3.3%) over 60 min, respectively, by walking in urban central areas (Fig. 4a). In comparison, suburban regions experience longer travel time to access essential necessities and services, with 41 (14.8%), 54 (19.6%), 49 (17.8%), and 79 (28.6%) countries falling within the 15-min, 15–30 min, 30–60 min, and over 60-min travel time brackets, respectively, by walking (Fig. 4b). Conversely, in rural areas, only 75 (27.2%) can access daily necessities and necessities within an acceptable travel time of <60 min, while the remaining 166 (60.1%)

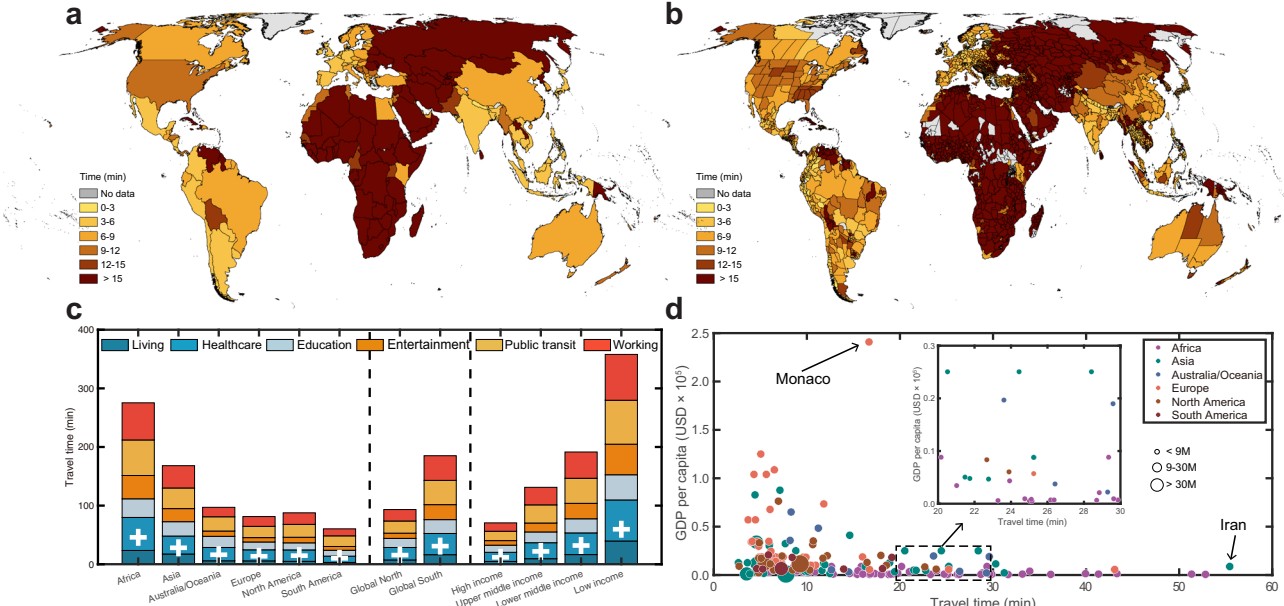

**Fig. 2 | Global patterns of mean travel time for urban residents to their living services and amenities with access to non-motorized transport.** Maps of mean travel time for urban residents' living services and amenities across country (**a**) and state (**b**) levels, where urban area is defined by global urban boundary (GUB) vector for 2020[33]. No data in **a**, **b** refers to countries or states without covering any GUB polygons. **c** Travel time for six types of living services and amenities by walk across different continent (Africa, Asia, Australia/Oceania, Europe, North America, and South America), urbanization (Global North and Global South), and income (high, upper middle, lower middle, and low incomes) contexts. The mean travel time for each continent, economic development, and income level are shown in each column as "+" symbol in white color. **d** The association between travel time and GDP per capita. Population data, extracted from the 100-m resolution WorldPop dataset, is categorized into three levels: <9 million, 9–30 million, and >30 million. A zoomed-in subplot, displaying travel times ranging between 20 and 30 min, is provided for detailed demonstration. Two countries with contrasting GDP and travel time, Monaco and Iran, are highlighted and labeled by name.

areas require at least 60 min of walking (Fig. 4c). Additionally, we found that disparities in travel time between urban centers and their surrounding areas are more pronounced in regions with lower economic development, regardless of transport modes (Fig. 4d and Supplementary Fig. 16d). For example, the average travel time difference by walk between urban and peri-urban areas is only 11.9 min for high-income countries, compared to 210.4 min in low-income countries. Likewise, these contrasts in travel time between urban and rural areas are intensified when moving from high-income to low-income countries.

## Discussion

One of the central aims for the United Nations' SDGs is to eliminate disparities in access to essential living necessities and services, thereby ensuring an inclusive, safe, sustainable, resilient, and high-quality life for all[2,22]. Existing efforts are predominantly focused on assessing and strategically optimizing infrastructure at a city-wide or street-block scale, employing the "x-minute city" model[36,37]. However, due to the absence of explicit spatial information on comprehensive infrastructure data and scalable accessibility measures, a scientific understanding of human access to essential daily necessities and services, and associated disparities remains elusive. To bridge this critical knowledge gap, we used the travel time as a proxy of accessibility[2,7] and contributed a higher spatial resolution global friction surface with fine-scale land cover type and road networks. By overlaying travel time with six major categories of essential infrastructures derived from global POI datasets, we mapped out pixel-level travel time worldwide, assessed progress and gaps towards the "x-minute city" model, and examined disparities of travel time across geographic, urbanization and socioeconomic spectrums.

Global essential infrastructure related to daily necessities and services, both in total numbers and per capita, are unevenly distributed across countries and socio-economic contexts (Fig. 1 and

Supplementary Fig. 2). High-income countries, which represent only 17.2% (1.35 billion) of the global population, occupy 30 million essential infrastructures, accounting for nearly 46.1% of the global total. In contrast, lower-middle- and low-income countries, comprising 48.9% (3.85 billion) of the global population, have access to just 11.7% (7.64 million) of these infrastructures. Moreover, individuals in high-income countries have access to ~3.70 living infrastructures per thousand residents, which is 8.8 times higher than the total number of those (0.42 living infrastructures per thousand residents) in lower-middle- and low-income countries. Notably, despite having a similar total number of infrastructures, residents in the Global South have only one-fifth of the infrastructure per capita compared to those in the Global North. These findings align with the general understanding that wealth dictates resource allocation[38–40], and further spatially elucidate the extent of global inequality in the distribution of essential infrastructures across different geographic and socioeconomic contexts. Beyond the number of essential infrastructures, their spatiotemporal interplay with population is more crucial, as this reflects whether resources are appropriately arranged and supplied to those who most need them[3,41,42]. To address this, we used travel time as an interpretable and comparable metric to estimate human access to those daily necessities and services and to inform associated socioeconomic disparities. Our statistics reveal that only 3.08 billion urban residents, or 62.8% of the global urban population, can reach their living necessities and services within a 15-min walk (Fig. 3). The population totals to 4.04 billion (82.5% of the global urban population) when the walking time threshold is increased to 30 min. These figures are far away from the target proposed in the 11th SDGs[21,22], indicating that much more effective efforts are needed to increase human proximity to essential living necessities and services.

A holistic understanding of travel time has important policy implications. First, travel time, as an interpretable and comparable

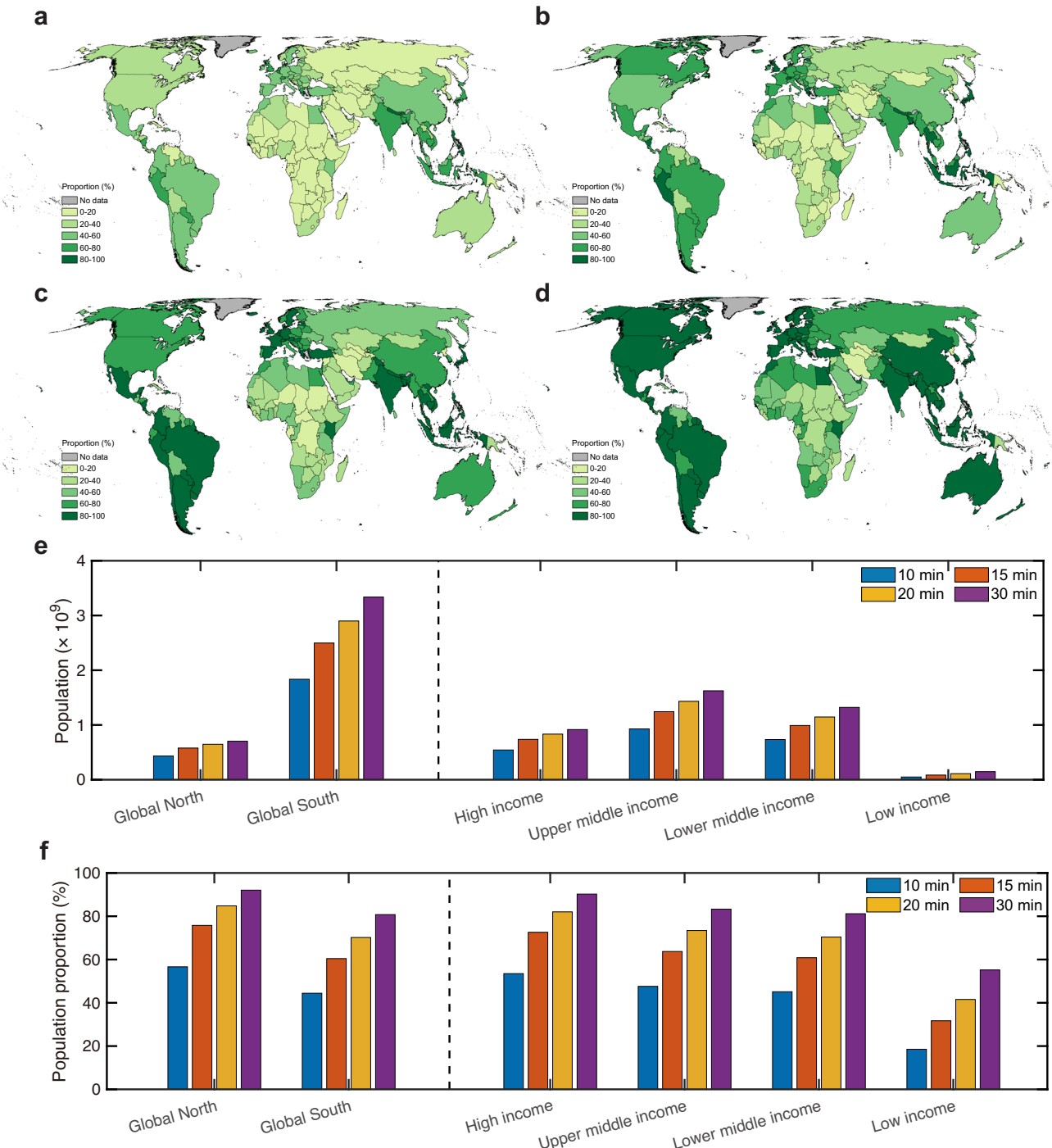

**Fig. 3 | Urban population's accessibility to living services and amenities under the "x-minute city" concept with non-motorized transport.** Maps of urban population proportion can reach their living services and amenities by a 10-min (**a**), 15-min (**b**), 20-min (**c**), and 30-min (**d**) walk. Absolute (**e**) and relative proportion (**f**) of urban population satisfy the "x-minute city" concept by walking across different urbanization (Global North and Global South) and income (high, upper middle, lower middle, and low incomes) contexts.

metric for measuring access to various resources and opportunities[2,7,11], effectively models the spatiotemporal interaction between human society and essential living necessities and services, considering both infrastructure supply and demand. For example, the relationship between travel time and the number of POIs per capita reveals different mechanisms affecting travel time costs across countries with varying levels of economic development (Supplementary Fig. 17). India, a country with high-density urban agglomerations where residents are close to essential services, has a comparable mean travel time estimate to that of Germany, which is well-serviced by such necessities. The mapped gridded travel time, with pixel-level explicit information on accessibility, provides valuable data to inform decisions by policymakers, designers, and other stakeholders helping improve strategies for the planning, placement, and management of infrastructures[43]. For example, this travel time information can be translated into actionable guidance to address urban-rural contrasts through two key steps: (1) identifying hotspot areas by overlaying travel-time maps with urban-rural population data (readily available to

local governments with demographic details) to pinpoint areas where high travel burdens disproportionately affect specific groups; and (2) developing targeted strategies for these hotspot areas through refining infrastructure types (e.g., enhancing necessary infrastructures or road networks in rural areas), optimizing spatial distributions (e.g., relocating or adding facilities based on demographic needs), and adopting tailor-tailed interventions to local demographic structures (e.g., increasing elderly-friendly services in aging communities). Second, while improving accessibility to essential services by reducing travel time may enhance living convenience, this could inadvertently decrease some opportunities for physical activity, a potential trade-off that warrants consideration. To mitigate this issue, urban planning and policy interventions should integrate strategies that encourage active lifestyles alongside infrastructure accessibility improvements. For instance, incorporating green infrastructure (e.g., urban parks and street trees) and walkable urban design (e.g., pedestrian-friendly pathways) can promote outdoor physical activity even in high-

accessibility areas. Additionally, expanding access to indoor fitness facilities (e.g., community gyms) could further support public health by providing alternative avenues for exercise. By adopting such measures, policymakers can ensure that accessibility enhancements do not come at the expense of population-level physical activity. Third, travel time to essential daily necessities and services is an integrated outcome of urbanization[44] and economic development[2]. The urbanization process enhances infrastructure investment as physical assets, boosting economic productivity, social prosperity, and promoting public health and well-being[45-47]. Disparities in travel time across the Global North and South, and along the urban-rural spectrum, offer insights into how urbanization correlates with infrastructure. In urban settings, the "15-minute city" model is increasingly popular among planners for its economic, environmental, and social benefits[21]. However, within this framework, access to and utilization of urban amenities vary across individuals and socioeconomic contexts. Interestingly, higher-income individuals, despite having access to high-quality local amenities[23], may opt for longer-distance travel[18], potentially increasing socioeconomic segregation in cities[48]. This nuanced implication of "15-minute" model globally warrants further exploration using realistic human mobility data. Furthermore, shortening travel time, especially for the motorized consumption with high-emission fuels (e.g., petroleum and natural gas), is critical for achieving net-zero emission targets, as the transport sector accounts for 30% of global energy use and around 40% of global carbon dioxide emissions[49]. Reducing travel distances, especially in motorized transport reliant peri-urban and rural areas, could significantly cut greenhouse gas emissions[50]. Lastly, the concept of time poverty—the chronic feeling of having too many tasks and not enough time to complete them—highlights an often-overlooked social issue exacerbated by ineffectively located daily necessities and services[1,51]. Addressing time poverty should be another integral point in sustainable and resilient urban planning.

Our study is subject to several levels of uncertainty. First, there are some limitations regarding data coverage and quality in global POI inventories and friction surface mapping (see "Method"). While we

**Table 1 | Global urban population and population fraction with an accessibility to living services and amenities under the "x-minute city" concept**

| Transportation | Time threshold | Urban population (in billions) | Urban population fraction |
|---|---|---|---|
| Non-motorized | 10-min | 2.27 | 46.3% |
| | 15-min | 3.08 | 62.8% |
| | 20-min | 3.55 | 72.5% |
| | 30-min | 4.04 | 82.5% |
| Motorized | 10-min | 4.41 | 90.0% |
| | 15-min | 4.58 | 93.6% |
| | 20-min | 4.66 | 95.3% |
| | 30-min | 4.73 | 96.7% |

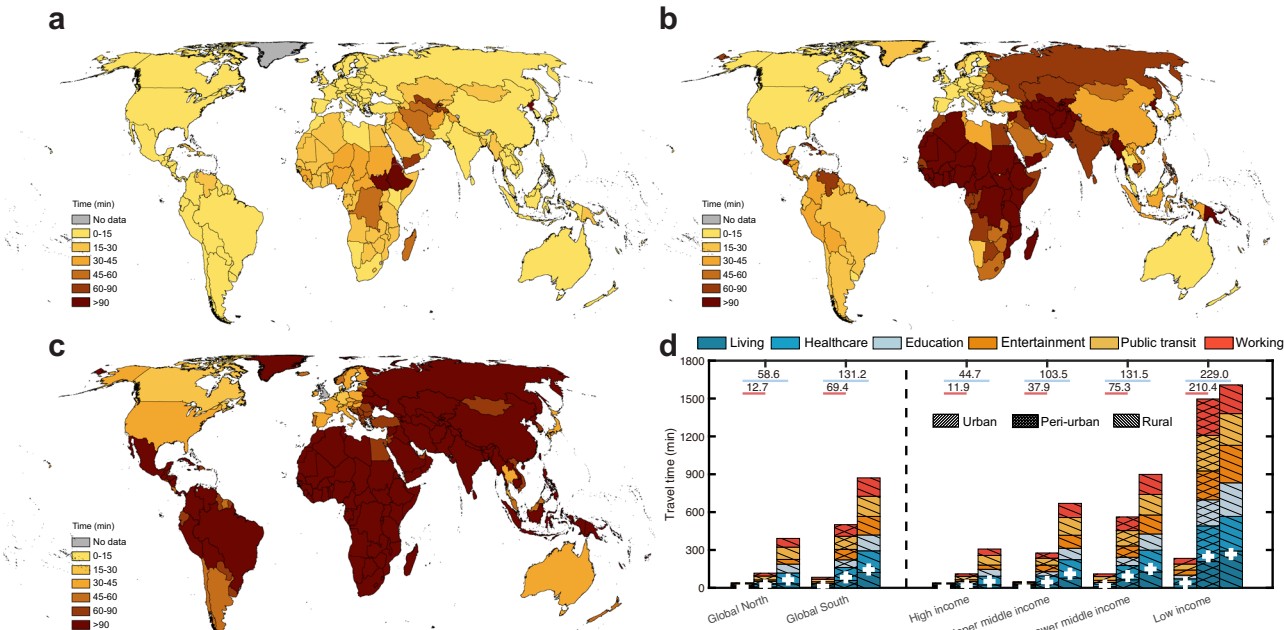

**Fig. 4 | Urban-rural contrasts in resident's travel time to living services and amenities with access to non-motorized transport.** Maps of mean travel time for urban residents' living services and amenities across urban (**a**), peri-urban (**b**), and rural (**c**) areas, with the associated boundary layers extracted from the degree of urbanization dataset in 2020 from the Global Human Settlement Layer (GHSL) project. **d** Travel time for six types of living services and amenities by walking across different urbanization (Global North and Global South) and income (high, upper middle, lower middle, and low incomes) contexts. The mean travel time for each social-economic development and income level are shown in each column as "+" symbol in white color. The mean travel time differences across urban-rural continuum degrees (peri-urban-urban and rural-urban) are also shown.

employed a unified six-category infrastructure classification framework for data normalization, the Overture and Amap POI datasets are compiled from diverse sources with differing methods and standards, introducing inconsistencies in data spatiotemporal coverage and quality and potentially affecting travel time calculations. Comparisons between two global POI datasets, including Overture (60 million records) and Foursquare (100 million records), reveal sample imbalances, particularly in low-income countries (Supplementary Fig. 18). Future efforts should prioritize finer-grained POI collection in these vulnerable regions. Friction surface mapping uncertainties arising from factors such as land cover, topography, and road networks further compound some uncertainties. The WorldCover product, while achieving 75% overall accuracy, shows reduced classification reliability in complex urban environments, particularly in shadowed zones beneath high-density structures. The ALOS DSM data for 2011 fails to capture subsequent topographical changes from natural processes (e.g., landslides) or human activities (e.g., mining operations), potentially affecting travel time estimates in dynamic landscapes. The Overture road network dataset may underestimate ghost roads in remote areas (e.g., Southeast Asia's forested regions[52]) and temporary roads in urban settings[53]. Integrating more multi-source, high-quality datasets are potential solutions for this critical issue. For instance, combining Overture data with local navigation platforms (e.g., Google Maps or Baidu) could improve detection of unlabeled roads in marginalized areas. Second, while travel time provides a useful proxy for potential physical accessibility (measured as minimum travel duration), actual travel behaviors are constrained by multiple factors, including individual transportation preferences (e.g., cost sensitivity and trip purposes), service quality (e.g., operational capacity and hours), and mobility constraints (e.g., weather conditions and traffic congestion). Our least-cost-path algorithm, which identifies the nearest resources for travel time estimation, assumes human prefer proximate services. However, this assumption may underestimate real accessibility, as actual utilization of daily necessities and services depends on the built environment, socioeconomic status, and personal preferences. Emerging human mobility data sources, such as mobile phone records (e.g., call detail records, Google Maps, and Strava) and social media check-ins (e.g., X and Foursquare), offer valuable individual-level insights into visitation patterns and service usage. Integrating these datasets with spatiotemporal interaction models that account for personal transportation behaviors and preferences could substantially improve our understanding of human access to essential services.

## Methods

### Global point of interest (POI) database for daily essential necessities and services

The global POIs dataset from Overture Maps (released on April, 2025) was used as a proxy measuring basic infrastructure of resident's daily life[31]. Overture provides nearly 60 million place records worldwide with associated confidence scores from a set of high-quality community-built data, including OpenStreetMap, high-resolution satellite/aerial imagery, government authoritative data, and newly open commercial data cube. This dataset allows us to quantify spatial locations of daily life infrastructures in a global context. As Overture's data coverage is rarely available in China due to policy restriction, we included an additional 12 million Chinese POIs from Amap to address this imbalanced sample issue[32]. Overture and Amap datasets provide around 2000 fine-granular categories (https://wiki.openstreetmap.org/wiki/Overture_categories). By combing these two POI datasets based on the first-level category schema, we excluded some non-related records (e.g., structure and geography) and organized the remaining into six major types ($N = 65,716,763$) to inform necessities and services in resident's daily life: living ($N = 41,394,922$), healthcare ($N = 4,261,898$), education ($N = 4,644,982$), entertainment

($N = 6,308,387$), public transit ($N = 4,486,765$), and working conditions ($N = 4,619,809$)[17,52] (Supplementary Tables 1–3).

### High-resolution global friction surface database

We adopted the friction surface approach[2,7,11], which can measure average speed at which human moves on Earth's landscape, to quantify residents' travel time to nearby service amenities. The friction surface method characterizes Earth surface as 2D regular grids/pixels and assigns movement speed to each pixel by considering a set of cost factors, including transportation infrastructure, water body, topography, and land cover. We created high-resolution global friction surface maps for non-motorized and motorized transports from three high-resolution products: Overture global road vector dataset (released on April, 2025) as a proxy of transportation infrastructure[31], 10-m-resolution European Space Agency (ESA) WorldCover product (version 100) for 2020 characterizing land cover[29], and 30-m-resolution ALOS World DSM (version 3.2) quantifying topography condition[30].

Road networks play a crucial role in friction surface mapping, as it largely determines human's speeds of movements across Earth surface. The Overture road network, primarily built on OpenStreetMap data, provides comprehensive global road coverage, with hierarchical classifications with 12 detailed sub-types (i.e., motorway, trunk, railroad, primary, secondary, tertiary, unclassified, residential, service, track, pedestrian, and others; Supplementary Fig. 19). We converted these individual road vectors into raster layers to characterize the friction surface. Given the absence of a global road width database and variations in worldwide standards, we applied a uniform empirical width of 10 m, aligning with the spatial resolution of WorldCover land cover, by buffering the Overture road vectors before rasterization. We modeled two baseline transportation modes: non-motorized (walking) and motorized transports, with actual travel behavior being a combination of both. Speed limits are critical for travel cost calculation. For motorized mode, we assigned country- and road-type-specific speed limits based on OpenStreetMap data, following a previous study[11]. Where country-specific data were unavailable, default global speeds were applied: motorways (105 km/h), trunks (80 km/h), railroads (30 km/h), primary (65 km/h), secondary (60 km/h), tertiary (55 km/h), unclassified (55 km/h), residential (40 km/h), service (30 km/h), tracks (40 km/h), pedestrian (25 km/h), and other roads (45 km/h) (Supplementary Fig. 20). For non-motorized mode, a constant walking speed of 5 km/h was adopted, consistent with the prior study[11]. Based on Overture road network, we generated a 1-km-resolution friction surface and compared it with Weiss's approach[11]. Our results show a significant lower friction values in human settlement areas, supporting that Overture provides denser road coverage (Supplementary Fig. 21).

For areas without transportation network, movement speeds were assumed to be determined by two factors: land cover types and topography. For land cover, we assigned baseline movement speeds to each category using the WorldCover product and a look-up-table approach. These speeds were initially derived from crowd-sourced data for 16 land cover types collected through an online survey[2]. We aggregated these survey-based speeds into 11 broader categories with the following average speeds: tree cover (3.06 km/h), shrubland (3.60 km/h), grassland (4.86 km/h), cropland (2.50 km/h), built-up areas (5.00 km/h), sparse vegetation (3.00 km/h), snow/ice (2.00 km/h), wetlands (2.00 km/h), mangroves (4.20 km/h), and moss (4.86 km/h). We excluded water body using WorldCover's water layer, as residents rarely use aquatic transportation for accessing daily services and amenities compared to land-based options.

Topography properties affect land-cover-dependent movement speeds through two key pathways. On one hand, higher elevations reduce atmospheric density and oxygen availability, decreasing maximal oxygen consumption ($VO_2$ max), a key determinant of cardiopulmonary performance, which consequently slows human movement

over rugged terrain[54]. Under a standard atmospheric condition, this elevation effect can be modeled as an adjustment factor (Eq. 1):

$$f_{elevation} = 1.016 \times e^{-0.0001072 \times elevation} \tag{1}$$

where $f_{elevation}$ is the adjustment factor of elevation (in meters).

On the other hand, slope steepness slows movement speeds by increasing energy expenditure and the frequency/duration of rest breaks[55]. We applied Tobler's Hiking Function to quantify the slope's effect on movement speeds (Eq. 2):

$$f_{slope} = 1.2 \times e^{-3.5 \times |\tan(0.01745 \times \alpha) + 0.05|} \tag{2}$$

where $\alpha$ is slope angle (in degrees) and $f_{slope}$ is adjustment factor of slope.

We derived the final friction surface by integrating elevation and slope adjustment factors with both land-cover-dependent and road-network-dependent movement speeds, applying a maximum composition approach. The global dataset was then resampled from its native 10-m resolution to 30-m resolution, balancing the need to capture urban environmental complexity with computational efficiency. To demonstrate the necessity of high-resolution friction mapping for local-scale analysis, we compared our results with Weiss et al.'s 1-km-resolution dataset (based on Open Street Map and Google roads, 500-m-resolution MCD12Q1 land cover, and 30-m-resolution Global Multi-resolution Terrain Elevation Data) (Supplementary Figs. 22–26). Our datasets capture much finer spatial details in both urban areas (Supplementary Figs. 23 and 25) and mountainous regions (Supplementary Fig. 26).

### Travel time mapping across different continental and socio-economic dimensions

Using the derived global friction surface raster and POI vector layers for six essential services and amenities (living, healthcare, education, entertainment, public transit, and employment), we applied the "cumulativeCost" function in Google Earth Engine[56] to map pixel-level travel time. This function, which implements Dijkstra's least-cost-path algorithm[57], measures accessibility as the time required to reach the nearest service from any given location. The "cumulativeCost" function requires three key inputs: (1) the friction surface (cost image), (2) a rasterized POI layer (source image), and (3) a maxDistance parameter defining the maximum search radius for path optimization. If the algorithm finds an accessible path within maxDistance, it returns the minimum travel time; otherwise, it returns zero. Through sensitivity analysis, we determined an optimal maxDistance of 30 km (equivalent to a 1000-pixel searing radius at 30-m resolution), balancing computational efficiency with accessibility measurement accuracy. To validate the travel time mapping algorithm, we conducted a comparative assessment using the Google Directions API as a reference (https://developers.google.com/maps/documentation/directions). First, we randomly selected 100 cities worldwide from the GUB dataset[34] and sampled 10 locations per city (Supplementary Fig. 27). For each city, we then constructed 45 origin-destination (O-D) pairs (i.e., $10 \times 9/2 = 45$, assuming the routes are reversible). Next, we computed travel time for all O-D pairs using both friction surface-based algorithm and the Google Directions API. Comparison results show strong consistency between the two methods for both non-motorized (walking) and motorized travel modes (with $R^2 = 0.94$ for non-motorized mode and $R^2 = 0.79$ for motorized mode; Supplementary Fig. 28), confirming the reliability of the proposed friction-based approach.

By integrating pixel-level travel time with gridded population distribution, we assessed accessibility for residents to living services and amenities across diverse socio-economic contexts. First, we mapped average travel time for urban residents and examined continental and socio-economic disparities. Second, based on travel time

maps, we evaluated the global applicability of the "x-minute city" concept by analysing accessibility under varying travel time thresholds. Lastly, we compared accessibility differences along the urban-rural continuum (i.e., urban, peri-urban, and rural areas). Within this context, we leveraged a set of multi-source spatial datasets, including the 2020 global GUB vector layer[34] for urban area extraction, the 1-km-resolution Global Human Settlement (GHS)-Settlement Model grid (GHS-SMOD) product for 2020[58] for rural-urban classifications, the 2015 Global Administrative Unit Layers (GAUL)[35] for extracting hierarchical administrative boundaries at state and country levels, the World Bank's GDP per capital (current USD) and income classification for 2022 (World Bank) for measuring social-economic status, and 100-m-resolution WorldPop dataset for 2020 with spatially explicit information on population density in 2D grids[33].

**Urban travel time mapping.** We adopted urban areas extracted from the GUB vector[34] as spatial units to map urban residents' travel times to essential services at state and country levels, subsequently quantifying socio-economic disparities. First, for each urban area, we generated 30-m-resolution travel time raster from global friction surfaces and POI datasets, resampled them to 100-m resolution for consistency with WorldPop population density data, and calculated population-weighted mean travel times. Second, we aggregated these travel time to state and country levels while incorporating mixed mobility scenarios with motorized fractions ranging from 0% to 100% in 10% increments. Third, we assessed accessibility disparities across: (1) continental regions (i.e., Asia, Africa, Australia/Oceania, Europe, North America, and South America), (2) global North/South divisions, and (3) World Bank income levels based on 2022 GNI per capita (low income: ≤1135 USD; lower-middle income: 1136–4465 USD; upper-middle income: 4466–13,845 USD; high income: ≥13,846 USD).

**"x-minute city" mapping.** The "x-minute city" urban planning aims to provide residents with access to daily essentials within an x-minute walk, bike ride, or public transit, with the "15-minute city" being the most well-known example[17]. We evaluated current status of the "x-minute city" concept by examining the mapped travel time data. To this end, we first identified urban areas where travel time to six critical infrastructures was shorter than the specified time thresholds (10, 15, 20, and 30 min) and counted the associated population and population percentage for these areas. We then compared the population numbers and percentages across urbanization spectrums (Global North and South) and income levels (high, upper-middle, lower-middle, and low income).

**Urban-rural differences in travel time.** We further investigated the disparity in travel time to essential infrastructure between urban and rural systems in two main steps. First, we selected three urbanization levels from the GHS-SMOD raster urban-rural continuum product by applying the "smod_code" layer as a mask: urban center (SMOD_code = 30), suburban or peri-urban (SMOD_code = 21), and rural (SMOD_code = 11) areas. Second, we mapped travel time across these three degrees of urbanization at country level. We then computed relative differences by comparing peri-urban (peri-urban minus urban) and rural (rural minus urban) travel time against urban centers as baseline, examining variations across urbanization spectrums (Global North/South) and income levels (high, upper-middle, lower-middle, low).

Spatial figures were generated across multiple scales (pixel, state, and country level) using ESRI ArcGIS 10.8. Data mapping and analysis were conducted using Google Earth Engine and MATLAB 2024b under non-commercial, research and education licenses.

## Data availability
Overture global datasets (point of interest (POI) and road network) are available at: https://overturemaps.org/; Amap POI data for China area

is available at: https://lbs.amap.com/; Foursquare global POI dataset is available at: https://location.foursquare.com/products/places/; 10-m-resolution European Space Agency (ESA) WorldCover product (version 100) for 2020 is available at: https://developers.google.com/earth-engine/datasets/catalog/ESA_WorldCover_v100; 30-m-resolution Advanced Land Observing Satellite (ALOS) World digital surface model (DSM) (version 3.2) is available at: https://developers.google.com/earth-engine/datasets/catalog/JAXA_ALOS_AW3D30_V3_2; Global urban boundary (GUB) vector layer for 2020 is available at: https://data-starcloud.pcl.ac.cn/zh; 1-km-resolution Global Human Settlement (GHS)-Settlement Model grid (GHS-SMOD) product for 2020 is available at: https://developers.google.com/earth-engine/datasets/catalog/JRC_GHSL_P2023A_GHS_SMOD; Global Administrative Unit Layers (GAUL) for 2015 is available at: https://developers.google.com/earth-engine/datasets/catalog/FAO_GAUL_2015_level0; 100-m-resolution WorldPop dataset for 2020 is available at: https://developers.google.com/earth-engine/datasets/catalog/WorldPop_GP_100m_pop; 1-km-resolution global friction surface for 2019 is available at: https://developers.google.com/earth-engine/datasets/catalog/Oxford_MAP_friction_surface_2019; Gross Domestic Product (GDP) per capital (current USD) for 2022 is available from World Bank at: https://data.worldbank.org/indicator/NY.GDP.MKTP.CD; Country-level income classification for 2022 is available from World Bank at: https://datahelpdesk.worldbank.org/knowledgebase/articles/906519-world-bank-country-and-lending-groups.

## Code availability

Script of travel time mapping with global friction surface image and cumulativeCost function is available from Google Earth Engine at: https://developers.google.com/earth-engine/apidocs/ee-image-cumulativecost.

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

## Acknowledgements
This study is supported by National Natural Science Foundation of China (42090015), the Research Grants Council of Hong Kong Strategic Topics Grant (STG2/P-705/24-R), Early Career Scheme (HKU27600222), and General Research Fund (HKU17601423), National Key Research and Development Program of China (2022YFB3903703), National Natural Science Foundation of China Young Scientists Fund (42201373), the Croucher Foundation (CAS22902/CAS22HU01), the Research Grants Council of the Hong Kong Special Administration Region (T35/710/20 R), The University of Hong Kong Start-up Fund for Professoriate Staff, Seed Fund for New Staff (103039010) and Basic Research (109002251), and The University of Hong Kong HKU-100 Scholars Fund.

## Author contributions
B.C. and P.G. conceived and designed the study. S.W. performed the main data analysis. S.W. and B.C. wrote the paper. J.A., A.N., F.D., C.L., and P.G. contributed to interpreting the results and reviewed and edited the paper.

## Competing interests
The authors declare no competing interests.
