## [Transparent Peer Review file · Nature Communications]

Measuring global human accessibility to essential daily necessities and services

Corresponding Author: Professor Bin Chen

Version 0:

Reviewer comments:

Reviewer #1

(Remarks to the Author)

The article delivers a groundbreaking global analysis of accessibility to essential services using an innovative 30-meter resolution friction surface to map travel times. This approach provides a detailed understanding of spatial inequalities, complemented by the compilation of a global points of interest (POI) database across six infrastructure categories. The findings highlight stark accessibility disparities across geographic, socio-economic, and urban-rural contexts, emphasizing the Global South's challenges in achieving equitable infrastructure access.

This study significantly contributes to urban planning, geography, and sustainable development, addressing critical gaps in global infrastructure accessibility. By linking findings to the "x-minute city" concept and aligning with SDG 11, it offers policymakers actionable insights for fostering inclusivity and sustainability.

Compared to prior studies constrained by scale and data limitations, this research employs advanced geospatial methods and a global perspective, enhancing its relevance with urban and rural gradients. The conclusions are well-supported by rigorous data analysis and visualization, though the friction surface model would benefit from validation through real-world datasets or case studies.

Areas for Further Improvement:

1. The authors should validate the friction surface and travel-time models with empirical data or real-world case studies to increase confidence in their findings.
2. Acknowledging and addressing potential biases in global POI datasets, particularly in underrepresented regions, would enhance the study's credibility.
3. More detailed and practical guidance for policymakers, particularly on addressing urban-rural accessibility disparities, would greatly enhance the study's real-world applicability. Additionally, while reducing travel distances is generally beneficial for improving access to services, it could inadvertently reduce physical activity for some individuals due to shorter walking distances. The potential trade-off between accessibility improvements and physical activity levels should be acknowledged and discussed by the authors to provide a more balanced perspective.
4. Exploring the use of open-source data alongside proprietary datasets would improve the reproducibility and accessibility of the research.

Minor edits:

1. Line 512: Wu adopted the friction ...
2. Some extra references which might be useful for the authors: <https://doi.org/10.1016/j.treng.2024.100244>
<https://doi.org/10.1080/17445647.2022.2141143>
<https://doi.org/10.1016/j.tra.2024.104354>
<https://doi.org/10.1016/j.tra.2024.104307>

Overall, this article is a significant contribution to global accessibility and infrastructure planning. With minor revisions to address these limitations, it is well-suited for publication and promises to make a lasting impact on the field.

Reviewer #2

(Remarks to the Author)

After reading the text, one can be preliminary tempted to say that the proposed analysis is characterized by scientific correctness (the methodology discussed, as well as the use of data - their sources are given, although this fact does not prejudge the ultimate usefulness). On the plus side, it should also be assessed that the authors are aware of their own research limitations - after all, the analysis covered the entire globe. And this is where my first doubt arises as to whether (how much) data limitations and methodological constraints affect the overall interpretation and generality of these findings in a broader cognitive context.

More specifically, global datasets of area, road infrastructure data, points (POI) have their imperfections and are not fully accurate in all regions as well as countries (generalizations can be seen/not seen due to their poor quality - on the maps presented, for example: US, Russia, Asian countries). GRIP data may omit roads in remote areas (how were temporary road/pedestrian routes actually included? Have they been included? How was information on paths to local roads from remote housing in the provinces used?). In addition, the global POI databases (Overture and Amap) come from different sources with different quality and standards, which may lead to inconsistencies - the explanations presented in the study are not enough for me individually. Please note that the absolute accuracy of some estimates may be questioned in practice, although overall trends and relationships may still be reliable.

The issue of modeling travel time has been addressed by many research teams around the world. Either detailed measurements were carried out and conclusions were formed based on them, or they relied on a variety of assumptions about the speed of movement, direction, etc., to get around under different conditions. The text treats coverage as well as terrain quite generally. This article also does not give enough information about the reliability in determining the class (types) of roads in the countries analyzed. In addition, the assumptions, although based on research and data, are a simplification of reality and may not reflect individual transportation behavior and preferences.

Travel duration is used as an indicator of accessibility, which is, of course, a common practice (by the way, I have conducted this kind of research myself). However, accessibility is a more complex concept and can depend on many other factors (cost, quality of service, opening hours, individual preferences, climatic conditions, time of year, etc.). Moreover, a short commute/arrival time alone does not guarantee full access. I also don't know how much consideration was given to the detailed directions of the road network, human travel behavior and differences in demographic structure? Is the "travel time" taken into account getting to the destination and returning home? Or just getting to the destination of the trip? Perhaps shuttle migrations should be discussed on a larger scale? The authors in the presented research did not pay attention to a very important feature of space - its resistance.

Despite my interest in the presented research, I am strongly disturbed in a secondary analysis of it by regional as well as national (and even global) averages. Arguably, they mask significant "local differences" and distinctive contexts. So, in practice, this may mean that the overall conclusions may not be fully "consistent" for a specific community (local/regional) or selected geographic area. I want to emphasize that I am not referring to false conclusions in the sense of violating the logic of the study, but rather to results that are subject to a certain margin of uncertainty and may require further research and verification in these specific contexts.

Knowing the conditions (including the results of research) of Europe or the United States, I am not able to accept the presented results (information) on individual maps. Their graphic quality is insufficient (analogously, descriptive analyses need improvement).

In the text, as well, there are many entries that are either a statement of the authors or have some source (without indicating it), for example: Globally, only 38.7% (2.6 31 billion people) and 50.7 % (3.4 billion people) of the population resides within a 15-minute and 32 30-minute walking distance of essential daily necessities and services, respectively. These results 33 highlight the urgent need to optimize strategies for planning, allocation, and management of 34 critical infrastructure to promote inclusive and sustainable development.

There are many uncertainties in this passage and statements that can be easily disproved by analyzing the sub-regional or regional scale in each country.

The literature review is not satisfactory, among other things, in terms of analyzing studies on cities of 15, 30 min, etc. A number of concepts and problems have not been satisfactorily explained, e.g. social justice, equitable access, etc. I do not develop this thread because it is the least of the problems of the reviewed text.

There are entries in the study, for me, at a very high level of generality ... almost a popular scientific level. As an example: Travel time contrast across the urban-rural continuum

We examined the difference in travel time between urban and rural areas, which represent the contrast in economic and environmental contexts and is significantly influenced by the geographical distribution of population density, transportation networks, and essential infrastructures (Fig. 4 and Extended Data Fig. 6). Our analysis reveals that among 276 countries, 117 (42.4%) have travel time to daily necessities and services within 15 minutes, 45 186 (16.3%) between 15-30 minutes, 20 (7.2%) between 30-60 minutes, and 7 (2.5%) over 60 minutes respectively, by walking in urban central areas (Fig. 4a). In comparison, suburban 188 regions experience longer travel time to access essential necessities and services, with 38 189 (13.8%), 57 (20.6%), 36 (13.0%), and 79 (28.6%) countries falling within the 15-minute, 15-30 190 minute, 30-60 minute, and over 60-minute travel time brackets, respectively, by walking (Fig. 191 4b). Conversely, in rural areas, only 72 (26.1%) can access daily necessities and necessities 192 within an acceptable travel time of <60 minutes, while the remaining 150 (54.4%) areas require at least 60 minutes of walking (Fig. 4c). Additionally, we found that disparities in travel time between urban centers and their surrounding areas are more pronounced in regions with lower economic development, regardless of transport modes (Fig. 4d and Extended Data Fig. 6d). For example, the average travel time difference between urban and peri-urban areas is only 19.4 minutes for high-income countries, compared to 196.7 minutes in low-income countries. Likewise, these contrasts in travel time between urban and rural areas are intensified when moving from high-income to low-income countries.

The text quoted above itself, as well as its graphic supplement, on the one hand, does not surprise - it was to be expected.

On the other hand, it presents an average value judgement that does not reflect (capture) regional conditions or relationships of interest to, among others, geographer, regionalist, urban planner, etc.

One last comment on the title: Measuring global human access to essential daily necessities and services. From a practical standpoint, everything is clear to the reader, but... Essential daily necessities and services are inherently local, and the title speaks of global access. The study is not concerned with measuring the existence or quality of local services themselves uniformly around the world. Instead, it focuses on measuring the spatial and temporal accessibility of these local services to people at the global level. In other words, it accounts for (measures) how easily (in terms of travel time) people in different parts of the world can reach their local resources and services needed to function.

In conclusion, the reviewed study provides a nice basis for conducting more extensive and effective research in the proposed subject area. It is obvious that one can use in a situation of data scarcity their interpolation - but not in this case. If you don't have complete data you don't have good research... we all eventually confront this problem in our professional lives.

Unfortunately, I do not recommend this text for publication. Currently, I do not see how it can be revised. I recommend limiting the research space to areas docked with complete and reliable information, and use data in addition: Light Detection and Ranging and/or Digital Elevation Model.

Version 1:

Reviewer comments:

Reviewer #1

(Remarks to the Author)

I would like to congratulate the authors on this important and timely piece of work. The revisions have significantly improved the clarity and overall quality of the manuscript. The study offers valuable insights into global accessibility patterns and highlights critical disparities that are highly relevant to both policy and research communities. I believe the paper makes a strong contribution to the field, and I am pleased to recommend it for publication in Nature Communications.

Reviewer #2

(Remarks to the Author)

The manuscript has been revised. The authors' responses are comprehensive and well-reasoned. They have demonstrated a deep understanding of the issues raised, making considerable efforts to improve the manuscript based on the comments received. Their approach, which includes detailed explanations and acknowledgement of remaining limitations, enhances the credibility of the study (the vast majority of my concerns have been addressed). However, it is clear that the final decision on the publication of the text remains with the Editor.

In my opinion, the text can be recommended for publication.

Response letter

> We appreciate four reviewers for their time spent in assessing this manuscript and for their thoughtful and valuable comments and suggestions, which are very helpful in improving our manuscript.

> Please see our point-by-point response to the reviewers' specific comments below.

Reviewer #1 (Remarks to the Author):

The article delivers a groundbreaking global analysis of accessibility to essential services using an innovative 30-meter resolution friction surface to map travel times. This approach provides a detailed understanding of spatial inequalities, complemented by the compilation of a global points of interest (POI) database across six infrastructure categories. The findings highlight stark accessibility disparities across geographic, socio-economic, and urban-rural contexts, emphasizing the Global South's challenges in achieving equitable infrastructure access.

This study significantly contributes to urban planning, geography, and sustainable development, addressing critical gaps in global infrastructure accessibility. By linking findings to the "x-minute city" concept and aligning with SDG 11, it offers policymakers actionable insights for fostering inclusivity and sustainability.

Compared to prior studies constrained by scale and data limitations, this research employs advanced geospatial methods and a global perspective, enhancing its relevance with urban and rural gradients. The conclusions are well-supported by rigorous data analysis and visualization, though the friction surface model would benefit from validation through real-world datasets or case studies.

> Thanks very much for your detailed reviews and positive comments regarding the implications, methodology, and study coverage, and summary of our key findings.

1) The authors should validate the friction surface and travel-time models with empirical data or real-world case studies to increase confidence in their findings.

> Thanks for this good suggestion. We substantially enhanced the spatial resolution of global friction surface dataset from 1 km to 30 meters by incorporating a more detailed road network layer and higher-resolution land cover and topography (DSM) data. We replaced the GRIP road network data with the Overture dataset, which is basically inherited from the latest OpenStreetMap (OSM). To demonstrate the advantage of this road dataset, we mapped a 1-km resolution global friction surface based on the Overture dataset (together with the 500-m-resolution MODIS MCD12Q1 land cover data and 90-m-resolution GMTED2010 DSM data) following Weiss's approach (Weiss et al. 2018, 2020) and compared it with Weiss's friction surface data. The Overture-based friction surface shows a consistent spatial pattern with Weiss's data (**Supplementary Fig. 21a** vs. **Supplementary Fig. 21b**), but significantly lower than the latter in human settlement areas (**Supplementary Fig. 21c**). Moreover, local examples demonstrate Overture is more spatially complete than the GRIP road network data (**Fig. R1**). These empirical comparisons confirm the reliability of our approach and suggest that the Overture road network is globally denser than those used in previous studies (Weiss et al. 2018, 2020).

Supplementary Fig. 21. Comparison of global friction surface for motorized transport between Overture-based (a) and Weiss's approach (b). **a.** Friction surface data that is generated from Overture road network (inherited from the latest OSM) together with 500-m-resolution MODIS MCD12Q1 land cover data and 90-m-resolution GMTED2010 DSM data. **b.** Weiss's friction surface that is generated from OSM and Google road for 2019 together with 500-m-resolution MODIS MCD12Q1 land cover data and 90-m-resolution GMTED2010 DSM data. **c.** Histogram distribution of the difference between the Overture-based and Weiss's friction surfaces, with statistics inserted in the plots.

Figure R1 redacted due to lack of 3rd Party Rights

Fig. R1. Comparison of road network coverage between GRIP (a, c) and Overture (b, d) in two urban areas. a-b. Fort Wayne, United States. c-d. Lusaka, Africa. Red curves represent road network overlaid on Google satellite basemaps.

Based on the Overture global road network data, together with the 10-m-resolution ESA WorldCover land cover data and 30-m-resolution ALOS DSM data, we further created a 30-m-resolution friction surface map. Compared with Weiss's friction surface, our data captures much finer spatial details over urban areas (**Supplementary Figs. 22-25**) and mountainous regions (**Supplementary Figs. 22, 24, and 26**).

Supplementary Fig. 22. Global maps of friction surface in non-motorized transport (walking mode) that was created by (a) this study, and (b) Weiss's approach. Friction surface in **a** was created from GRIP road network, 10-m-resolution WorldCover land cover, and 30-m-resolution ALOS DSM datasets and resampled to 1-km resolution for visualization. Friction surface in **b** was created from OSM and Google Road networks, 500-m-resolution MCD12Q1 land cover, and 30-m-resolution GMTED2010 DSM datasets with a final 1-km data resolution.

Supplementary Fig. 23 redacted due to lack of 3rd Party Rights

Supplementary Fig. 23. Local examples of friction surface for non-motorized (walking) transport generated by (b and e) this study and (c and f) Weiss's approach over (a-c) Chicago, United States and (d-f) Shanghai, China. a and d, Google satellite image; b and e, friction surface of this study; c and f, friction surface of Weiss's approach.

Supplementary Fig. 24. Global maps of friction surface for the motorized transport that was created by (a) this study, and (b) Weiss's approach. a. Friction surface that was created from GRIP road network, 10-m-resolution WorldCover land cover, and 30-m-resolution ALOS DSM datasets and resampled to 1-km resolution for visualization. **b.** Friction surface that was created from OSM and Google Road networks, 500-m-resolution MCD12Q1 land cover, and 30-m-resolution GMTED2010 DSM datasets with a final 1-km data resolution.

Supplementary Fig. 25 redacted due to lack of 3rd Party Rights

Supplementary Fig. 25. Local examples of friction surface for motorized transport generated by (b and e) this study and (c and f) Weiss's approach over (a-c) Chicago, United States and (d-f) Shanghai, China. a and d, Google satellite image; **b and e,** friction surface of this study; **c and f,** friction surface of Weiss's approach.

Supplementary Fig. 26 redacted due to lack of 3rd Party Rights

Supplementary Fig. 26. Local examples of friction surface over two mountainous areas. a-c. Tibetan plateau Mountain. **d-f.** Andres Mountain. **a** and **d.** Google satellite image. **b** and **e.** Friction surface map created in this study. **c** and **f.** Weiss's friction surface map.

To validate the accuracy of our travel time algorithm, we conducted a comparative assessment using the Google Directions API as a reference (<https://developers.google.com/maps/documentation/directions>). First, we randomly selected 100 cities worldwide from the Global Urban Boundary (GUB) dataset (Li et al., 2020) and generated 10 location points per city (Supplementary Figs. X). For each city, we then constructed 45 origin-destination (O-D) pairs (i.e., $10 \times 9/2 = 45$), assuming that the routes are reversible. Next, we computed travel times for all O-D pairs using both our algorithm and the Google Directions API. The results show strong consistency between the two methods for both non-motorized (walking) and motorized travel modes (with $R^2 = 0.94$ for non-motorized mode and $R^2 = 0.79$ for motorized mode; **Supplementary Figs. 27-28**), demonstrating the reliability of our approach.

Supplementary Fig. 27 redacted due to lack of 3rd Party Rights

Supplementary Fig. 27. Global sample cities for travel time validation. a. Spatial distribution of 100 sample cities (red points). **b-g.** Local examples of six sample cities with 10 validation points (red pentagrams) for each, which are bounded by google urban boundary (GUB) dataset and overlaid on Google satellite basemaps.

Supplementary Fig. 28. Accuracy validation of travel time for (a) non-motorized and (b) motorized modes. Google travel time is calculated from Google Directions API and our travel time is calculated from the friction surface with the “least-cost-path” algorithm.

We have updated the methodology of global friction surface generation (Pages 14-15, Lines 466-486), and included the empirical validations of our friction surface (Pages 16, Lines 533-538) and travel time into the revised manuscript (Pages 16, Lines 551-561), which is duplicated as follows:

(Pages 14-15, Lines 466-486) “Road networks play a crucial role in friction surface mapping, as it largely determines human’s speeds of movements across Earth surface. The Overture road network, primarily built on OpenStreetMap data, provides the most comprehensive global road coverage, with hierarchical classifications with 12 detailed sub-types (i.e., motorway, trunk, railroad, primary,

secondary, tertiary, unclassified, residential, service, track, pedestrian, and others; **Supplementary Fig 19**). We converted these individual road vectors into raster layers to characterize the friction surface. Given the absence of a global road width database and variations in worldwide standards, we applied a uniform empirical width of 10 meters, aligning with the spatial resolution of WorldCover land cover, by buffering the Overture road vectors before rasterization. We modelled two baseline transportation modes: non-motorized (walking) and motorized transports, with actual travel behaviour being a combination of both. Speed limits are critical for travel cost calculation. For motorized mode, we assigned country- and road-type-specific speed limits based on OpenStreetMap data, following a previous study⁹. Where country-specific data were unavailable, default global speeds were applied: motorways (105 km/h), trunks (80 km/h), railroads (30 km/h), primary (65 km/h), secondary (60 km/h), tertiary (55 km/h), unclassified (55 km/h), residential (40 km/h), service (30 km/h), tracks (40 km/h), pedestrian (25 km/h), and other roads (45 km/h) (**Supplementary Fig. 20**). For non-motorized mode, a constant walking speed of 5 km/h was adopted, consistent with the prior study⁹. Based on Overture road network, we generated a 1-km-resolution friction surface and compared it with Weiss's approach⁹. Our results show a significant lower friction values in human settlement areas, supporting that Overture provides denser road coverage (**Supplementary Fig. 21**)."

(Pages 16, Lines 533-538) "To demonstrate the necessity of high-resolution friction mapping for local-scale analysis, we compared our results with Weiss et al.'s 1-km-resolution dataset (based on Open Street Map and Google roads, 500-m-resolution MCD12Q1 land cover, and 30-m-resolution Global Multi-resolution Terrain Elevation Data) (**Supplementary Figs. 22-26**). Our datasets capture much finer spatial details in both urban areas (**Supplementary Figs. 23 and 25**) and mountainous regions (**Supplementary Fig. 26**)."

(Pages 16, Lines 551-561) "To validate the travel time mapping algorithm, we conducted a comparative assessment using the Google Directions API as a reference (<https://developers.google.com/maps/documentation/directions>). First, we randomly selected 100 cities worldwide from the Global Urban Boundary (GUB) dataset³⁵ and sampled 10 locations per city (**Supplementary Fig. 27**). For each city, we then constructed 45 origin-destination (O-D) pairs (i.e., $10 \times 9/2 = 45$), assuming the routes are reversible. Next, we computed travel times for all O-D pairs using both friction surface-based algorithm and the Google Directions API. Comparison results show strong consistency between the two methods for both non-motorized (walking) and motorized travel modes (with $R^2 = 0.94$ for non-motorized mode and $R^2 = 0.79$ for motorized mode; **Supplementary Figs. 28**), confirming the reliability of the proposed friction-based approach."

References:

Weiss, D. J., Nelson, A., Vargas-Ruiz, C. A., Gligorić, K., Bavadekar, S., Gabrilovich, E., ... & Gething, P. W. (2020). Global maps of travel time to healthcare facilities. *Nature Medicine*, 26(12), 1835-1838

Weiss, D. J., Nelson, A., Gibson, H. S., Temperley, W., Peedell, S., Lieber, A., ... & Gething, P. W. (2018). A global map of travel time to cities to assess inequalities in accessibility in 2015. *Nature*, 553(7688), 333-336.

2) Acknowledging and addressing potential biases in global POI datasets, particularly in underrepresented regions, would enhance the study's credibility.

> Thanks for this good suggestion. We agree that the spatial coverage and data quality of global POI datasets are crucial to travel time mapping. To increase spatial coverage of global POI, we updated the Overture POI data with the latest version (<https://overturemaps.org/>), increasing the total records of global POI inventory from 58,027,024 to 65,716,763. We then assessed the accuracy of Overture + Amap POI data with reference from Foursquare's global (Foursquare + Amap, <https://location.foursquare.com/products/places/>) POI with a total record of 90,805,597, which shows closely spatial distribution of POI density (**Supplementary Fig. 18**). Although Foursquare + Amap provides more POI records, we still believe that Overture + Amap is the ideal POI inventory for global travel time mapping for two major reasons. First, when diving into the spatial details, we found that the Overture + Amap POI has a much more balanced data coverage (**Supplementary Fig. 18**), especially for the developing countries or regions which are usually underrepresented (Herfort et al. 2023). Second, Overture provides an accuracy confidence layer (with value from 0 to 1) to indicate the data quality of POI, helping us validate the result robustness under different uncertainty levels with sensitivity analysis (**Supplementary Figs.5-6**).

Supplementary Fig. 18. Comparison of POI densities (total POI number per km) between Foursquare + Amap (a, c-f) and Overture + Amap (b, h-l). a-b. Global maps of Foursquare + Amap (a) and Overture + Amap (b) POI densities. c-l. Local distributions of POI densities across five countries: United State (c, h), Germany (d, i), India (e, j), Bolivia (f, k), and Uganda (g, l). m. Scatter plot of Foursquare + Amap and Overture + Amap POI densities.

Supplementary Fig. 5. Global distribution of Overture + Amap POI density under different confidence levels. a. Confidence level of 0.70 (total POI records $N = 49,074,325$). **b.** Confidence level of 0.80 ($N = 42,591,715$). **c.** Confidence level of 0.90 ($N = 34,413,184$).

Supplementary Fig. 6. Sensitivity analysis of travel time for six types of living services and amenities using with different confidences levels of global POI. (a, b) Confidence level of 0.7, (c, d) Confidence level of 0.8, and (e, f) Confidence level of 0.9. a, c, and e, Travel time by non-motorized mode across different continent (Africa, Asia, Australia/Oceania, Europe, North America, and South America indicated by slight blue background), urbanization (Global North and Global South indicated by slight pink background), and income (high, upper middle, lower middle, and low incomes indicated by slight yellow background) contexts. b, d, and f, Travel time by motorized mode across different continent (Africa, Asia, Australia/Oceania, Europe, North America, and South America indicated by slight blue background), urbanization (Global North and Global South indicated by slight pink background), and income (high, upper middle, lower middle, and low incomes indicated by slight yellow background) contexts. The mean travel time for each social-economic development and income level are shown in each column as “+” symbol in white colour.

Despite the high quality of global Overture + Amap POI data, it is still inevitable to suffer from data scarcity in some specific areas, such as low or middle-income countries due to the lack of individual volunteer mapping or organized humanitarian mapping efforts (Herfort et al. 2023). Temporal change of POI is another important uncertainty factor but has not yet been fully addressed. Most POI sources only provide baseline coverage for specific years or limited temporal coverage, making it difficult to explore the spatiotemporal dynamics of daily necessities and services and the associated underlying socioeconomic drivers (Pandey et al. 2025).

To make these points clear, we have updated the methodology regarding POI dataset update (Page

14, Lines 446-451), and sensitivity analysis with different confidence layers (Page 5, Lines 169-171), and also discussed the potential uncertainties regarding the quality and coverage of global POI data in the revised manuscript (Page 8, Lines 312-320), which is duplicated as follows:

(Page 14, Lines 446-451) “By combing these two POI datasets based on the first-level category schema, we excluded some non-related records (e.g., structure and geography) and organized the remaining into six major types (N = 65,716,763) to inform necessities and services in resident’s daily life: living (N = 41,394,922), healthcare (N = 4,261,898), education (N = 4,644,982), entertainment (N = 6,308,387), public transit (N = 4,486,765), and working conditions (N = 4,619,809)^{16,55} (**Supplementary Tables 1-3**).”

(Page 5, Lines 169-171) “These inequities in travel time are consistent across different quality levels of global POI inventories (**Supplementary Figs. 5 and 6**).”

(Page 8, Lines 312-320) “Our study is subject to several levels of uncertainty. First, there are some limitations regarding data coverage and quality in global POI inventories and friction surface mapping (**see Method**). While we employed a unified six-category infrastructure classification framework for data normalization, the Overture and Amap POI datasets are compiled from diverse sources with differing methods and standards, introducing inconsistencies in data coverage and quality and potentially affecting travel time calculations. Comparisons between two global POI datasets, including Overture (60 million records) and Foursquare (100 million records), reveal sample imbalances, particularly in less developed countries (**Supplementary Fig. 18**). Future efforts should prioritize finer-grained POI collection in these vulnerable regions”

References:

Herfort, B., Lautenbach, S., Porto de Albuquerque, J., Anderson, J., & Zipf, A. (2023). A spatio-temporal analysis investigating completeness and inequalities of global urban building data in OpenStreetMap. *Nature Communications*, 14(1), 3985.

Pandey, B., Brelford, C., & Seto, K. C. (2025). Rising infrastructure inequalities accompany urbanization and economic development. *Nature Communications*, 16(1), 1193.

3) More detailed and practical guidance for policymakers, particularly on addressing urban-rural accessibility disparities, would greatly enhance the study's real-world applicability. Additionally, while reducing travel distances is generally beneficial for improving access to services, it could inadvertently reduce physical activity for some individuals due to shorter walking distances. The potential trade-off between accessibility improvements and physical activity levels should be acknowledged and discussed by the authors to provide a more balanced perspective.

> Thanks for these insightful suggestions to enhance the real-world applicability of our study and address potential trade-offs in policymaking. Our proposed mapping methodology and dataset provide spatially explicit travel-time metrics to daily necessities and services. We can translate these findings into actionable guidance through two key steps: 1) identifying hotspot areas by overlaying travel-time maps with urban-rural population data (readily available to local governments with demographic details) to pinpoint areas where high travel burdens disproportionately affect specific groups; and 2)

developing targeted strategies for these hotspot areas through refining infrastructure types (e.g., enhancing necessary infrastructures or road networks in rural areas), optimizing spatial distributions (e.g., relocating or adding facilities based on demographic needs), and adopting tailor-tailed interventions to local demographic structures (e.g., increasing elderly-friendly services in aging communities).

It should clarify that this study maps human accessibility to essential necessities and services, such as medicine and education resources, which greatly supports human's daily lives beyond outdoor physical activities. While improving accessibility to essential services by reducing travel distances may enhance convenience, we acknowledged that this could inadvertently decrease opportunities for physical activity—a potential trade-off that warrants consideration. To mitigate this issue, urban planning and policy interventions should integrate strategies that encourage active lifestyles alongside accessibility improvements. For instance, incorporating green infrastructure (e.g., urban parks and street trees) and walkable urban design (e.g., pedestrian-friendly pathways) can promote outdoor physical activity even in high-accessibility areas. Additionally, expanding access to indoor fitness facilities (e.g., community gyms) could further support public health by providing alternative avenues for exercise. By adopting such measures, policymakers can ensure that accessibility enhancements do not come at the expense of population-level physical activity.

To make these points clear, we discussed the practical guidance and some potential solutions to balance the trade-off between infrastructure accessibility improvements and outdoor physical exercise reductions in the revised manuscript (Page 7, Lines 272-291), which is duplicated as follows:

“For example, this travel time information can be translated into actionable guidance to address urban-rural contrasts through two key steps: 1) identifying hotspot areas by overlaying travel-time maps with urban-rural population data (readily available to local governments with demographic details) to pinpoint areas where high travel burdens disproportionately affect specific groups; and 2) developing targeted strategies for these hotspot areas through refining infrastructure types (e.g., enhancing necessary infrastructures or road networks in rural areas), optimizing spatial distributions (e.g., relocating or adding facilities based on demographic needs), and adopting tailor-tailed interventions to local demographic structures (e.g., increasing elderly-friendly services in aging communities). Second, while improving accessibility to essential services by reducing travel time may enhance living convenience, this could inadvertently decrease some opportunities for physical activity, a potential trade-off that warrants consideration. To mitigate this issue, urban planning and policy interventions should integrate strategies that encourage active lifestyles alongside accessibility improvements. For instance, incorporating green infrastructure (e.g., urban parks and street trees) and walkable urban design (e.g., pedestrian-friendly pathways) can promote outdoor physical activity even in high-accessibility areas. Additionally, expanding access to indoor fitness facilities (e.g., community gyms) could further support public health by providing alternative avenues for exercise. By adopting such measures, policymakers can ensure that accessibility enhancements do not come at the expense of population-level physical activity.”

4) Exploring the use of open-source data alongside proprietary datasets would improve the reproducibility and accessibility of the research.

> Thanks for this good suggestion. Both two POI dataset, including Overture (non-China areas) and Amap (China area) are freely available for the public. We add the data websites of these two datasets in the data availability section, which is duplicated as follows:

“Overture global datasets (point of interest (POI) and road network) are available at: <https://overturemaps.org/>; Amap POI data for China area is available at: <https://lbs.amap.com/>.”

5) Line 512: Wu adopted the friction ...

> Thanks for pointing out this typo, we have corrected accordingly (Page 14, Lines 454-455), which is duplicated as follows:

“We adopted the friction surface approach^{2,6,9}, which can measure average speed at which human moves on Earth’s landscape, to quantify residents’ travel time to nearby service amenities.”

6) Some extra references which might be useful for the authors:

<https://doi.org/10.1016/j.treng.2024.100244>

<https://doi.org/10.1080/17445647.2022.2141143>

<https://doi.org/10.1016/j.tra.2024.104354>

<https://doi.org/10.1016/j.tra.2024.104307>

> Thanks for these relevant literatures regarding “*x-minute*” city mapping and modeling, we have involved them in our introduction sections (Page 3, Lines 76-85), which is duplicated as follows:

“A central focus of the “x-minute city” concept is developing holistic methods or metrics to evaluate accessibility to essential services, considering factors such as service availability, road networks, active mobility options, and socioeconomic diversity²³⁻²⁴. Location-based approaches—such as proximity-based measures and the two-step floating catchment area (2SFCA) method—are widely used to assess accessibility, quantifying service availability within specific travel distances or times²⁵. More advanced models, such as spatial-temporal interaction frameworks, incorporate additional dimensions like service capability, attractiveness, user preferences, and human mobility patterns to better reflect actual usage of amenities¹⁶. Among these metrics, proximity-based measures (e.g., accessibility or proximity) remain the most common due to their simplicity and efficiency.”

7) Overall, this article is a significant contribution to global accessibility and infrastructure planning. With minor revisions to address these limitations, it is well-suited for publication and promises to make a lasting impact on the field.

> Thank you for your encouragement on the broad implications of our study.

Reviewer #2 (Remarks to the Author):

After reading the text, one can be preliminary tempted to say that the proposed analysis is characterized by scientific correctness (the methodology discussed, as well as the use of data - their sources are given, although this fact does not prejudice the ultimate usefulness). On the plus side, it should also be assessed that the authors are aware of their own research limitations - after all, the analysis covered the entire globe. And this is where my first doubt arises as to whether (how much) data limitations and methodological constraints affect the overall interpretation and generality of these findings in a broader cognitive context.

> We sincerely appreciate the reviewer's recognition of our study's scientific rigor and their thoughtful concerns regarding data, methodology, and result interpretation. In response, we have implemented several improvements to strengthen our work:

(1) Data quality: we have enhanced data quality and coverage by incorporating the denser global POI and road network datasets (i.e., latest Overture dataset released on April 2025) and compared these datasets with other alternatives (i.e., Foursquare global POI and OSM + Google road datasets) to demonstrate their superiority (**Supplementary Figs. 18, 21-25**). Besides, we clarified the usage of high-resolution global digital surface model (DSM) product (i.e., 30-m-resolution ALOS DSM) to measure topographic effects on friction surface generation (**Supplementary Fig. 16**).

(2) Methodology development: we rigorously validated our travel time mapping algorithm using the benchmarks from Google Directions API (**Supplementary Figs. 27-28**), and added local examples of travel time mapping over four cities (i.e., Berlin, Paris, Chicago, and Washington; **Supplementary Fig. 8-15**).

(3) Result robustness: we confirmed the robustness of our key findings by conducting comprehensive sensitivity analysis with different confidence thresholds of the global POI dataset (**Supplementary Figs. 5-6**), and included statistic results of global urban population meet the "x-minute city" concept (**Table 1**)

We clarified these improvements together with both global and local-level analysis in the revised manuscript. Please see our point-by-point responses below.

1) More specifically, global datasets of area, road infrastructure data, points (POI) have their imperfections and are not fully accurate in all regions as well as countries (generalizations can be seen/not seen due to their poor quality - on the maps presented, for example: US, Russia, Asian countries). GRIP data may omit roads in remote areas (how were temporary road/pedestrian routes actually included? Have they been included? How was information on paths to local roads from remote housing in the provinces used?). In addition, the global POI databases (Overture and Amap) come from different sources with different quality and standards, which may lead to inconsistencies - the explanations presented in the study are not enough for me individually. Please note that the absolute accuracy of some estimates may be questioned in practice, although overall trends and relationships may still be reliable.

> Thank you for these valuable comments regarding the spatial coverage, quality, and standardization

of global POI and road network data. We would like to clarify that our study utilizes the most comprehensive and representative POI and road network datasets currently available to the research community. The two POI datasets: Overture (which is built on OSM and other high-quality open-source data, majorly covering global area beyond China; this data has been updated for the latest version on April 2025) and Amap (majorly covering China) are currently two most accurate dataset across the community and serve as reference data source for many other accessibility-related studies at local and global spatial contexts. To validate the spatial coverage of our global POI data (Overture + Amap), we used another global commercial POI dataset from the Foursquare company (Foursquare + Amap, <https://location.foursquare.com/products/places/>) as references, which shows our POI data is spatially consistent with Foursquare's data (**Supplementary Fig. 18**), but has a superior data coverage than the latter over low-income or middle-income countries (e.g., India, Bolivia, and Uganda, **Supplementary Fig. 18e-g** and **Supplementary Fig. 18j-l**). As Overture and Foursquare are currently the two most representative global POI datasets, and given our validation results, we are confident in the appropriateness of the Overture POI data for global-scale travel time mapping.

Supplementary Fig. 18. Comparison of POI densities (total POI number per km) between Foursquare + Amap (a, c-f) and Overture + Amap (b, h-l). a-b. Global maps of Foursquare + Amap (a) and Overture + Amap (b) POI densities. c-l. Local distributions of POI densities across five example countries: United State (c and h), Germany (d and i), India (e and j), Bolivia (f and k), and Uganda (g and l). m. Scatter plot of Foursquare + Amap and Overture + Amap POI densities.

Due to regulatory restrictions, the Overture POI dataset exhibits limited coverage in China (approximately 0.87 million records, representing just 6.8% of the total 12.64 million records analyzed). To overcome this spatial data gap, we integrated high-quality POI data from Amap, which is widely recognized as the most authoritative POI source for China. To ensure compatible integration of these two datasets, we implemented a rigorous harmonization process by using a unified classification framework that categorizes all POIs into six standardized classes (**Tables S1-S2**) and maintaining

consistent data quality standards throughout the integration process.

Table S1. Classification scheme of six major necessities and services in residents' daily life for the Overture global place of interest (POI) database.

Class	Overture scheme
Living	Eat and drink
Living	Accommodation
Living	Beauty and spa
Living	Financial service
Living	Retail
Living	Home service
Living	Professional services
Healthcare	Health and medical
Education	Education
Entertainment	Arts and entertainment
Entertainment	Attractions and activities
Entertainment	Active life
Public transit	Travel
Working	Business to business

Table S2. Classification scheme of six major necessities and services in residents' daily life for the Amap Chinese place of interest (POI) database.

Class	Amap scheme	Amap scheme (Chinese)
Living	Accommodation services	住宿服务
Living	Public utilities	公共设施
Living	Business residence	商务住宅
Living	Domestic services	生活服务
Living	Shopping services	购物服务
Living	Financial insurance services	金融保险服务
Living	Catering services	餐饮服务
Healthcare	Health care services	医疗保健服务
Education	Science, education and cultural services	科教文化服务
Entertainment	Sports and leisure services	体育休闲服务
Entertainment	Famous tourist sites	风景名胜
Public transit	Transportation facilities services	交通设施服务
Public transit	Motorcycle services	摩托车服务
Public transit	Car services	汽车服务
Public transit	Car repair	汽车维修
Public transit	Car sale	汽车销售
Public transit	Access facilities	通行设施
Public transit	Road ancillary facilities	道路附属设施
Working	Company	公司企业

Regarding the global road network data, we upgraded our road network data source from GRIP (based on pre-2018 OSM data) to the more recent Overture road dataset (built on April 2025 OSM

data). This update provides significant advantages with finer classification (with 12 levels of road types: motorway, trunk, railroad, primary, secondary, tertiary, unclassified, residential, service, track, pedestrian, and others) and more comprehensive spatial coverages (comparisons between Overture and OSM + Google road hybrid network in **Supplementary Fig. 21**; comparisons between Overture and GRIP road network in **Fig. R1**).

We believe in temporary roads/pedestrian routes or remote areas have a minor impact on our analysis of global infrastructure accessibility for two reasons. On the one hand, according to the statistics of global OSM data (the main resource of Overture data), temporal roads account for only 0.36% of global road network (see the status of highway=construction in the right panel: <https://wiki.openstreetmap.org/wiki/Tag:highway%3Dconstruction>), suggesting a negligible contribution from this uncertainty factor. Second, remote areas typically have low residing population density and then the exclusion of these areas rarely affects global accessibility comparisons across geographic, socio-economic, and urban-rural contexts.

Supplementary Fig. 21. Comparison of global friction surface for motorized transport between Overture-based (a) and Weiss's approach (b). a. Friction surface data that is generated from Overture road network (inherited from the latest OSM) together with 500-m-resolution MODIS MCD12Q1 land cover data and 90-m-resolution GMTED2010 DSM data. b. Weiss's friction surface that is generated from OSM and Google road for 2019 together with 500-m-resolution MODIS MCD12Q1 land cover data and 90-m-resolution GMTED2010 DSM data. c. Histogram distribution of the difference between the Overture-based and Weiss's friction surfaces, with statistics inserted in the plots.

Figure R1 redacted due to lack of 3rd Party Rights

Fig. R1. Comparison of road network coverage between GRIP (a, c) and Overture (b, d) in two urban areas. a-b. Fort Wayne, United States. **c-d.** Lusaka, Africa. Red curves represent road network overlaid on Google satellite basemaps.

Despite the comprehensive coverage of our global datasets, we acknowledge that these datasets might be less representative over some specific local contexts, such as spatiotemporal evolutions of POI distribution and road network expansion. We have enriched our global POI and road network by using the latest products (Page 14, Lines 737-738) and discussed the data limitations with potential solutions in the revised manuscript (Page 8, Lines 312-331), which is duplicated as follows:

(Page 14, Lines 737-738) *“The global POIs dataset from Overture Maps (released on April, 2025) was used as a proxy measuring basic infrastructure of resident’s daily life³⁰.”*

(Page 8, Lines 312-331) *“Our study is subject to several levels of uncertainty. First, there are some limitations regarding data coverage and quality in global POI inventories and friction surface mapping (see **Method**). While we employed a unified six-category infrastructure classification framework for data normalization, the Overture and Amap POI datasets are compiled from diverse sources with differing methods and standards, introducing inconsistencies in data coverage and quality and potentially affecting travel time calculations. Comparisons between two global POI datasets, including Overture (60 million records) and Foursquare (100 million records), reveal sample imbalances, particularly in less developed countries (**Supplementary Fig. 18**). Future efforts should prioritize finer-grained POI collection in these vulnerable regions. Friction surface mapping uncertainties arising from factors such as land cover, topography, and road networks, further compound some uncertainties. The WorldCover product, while achieving 75% overall accuracy, shows reduced classification reliability in complex urban environments, particularly in shadowed zones beneath high-density*

structures. The ALOS DSM data for 2011 fails to capture subsequent topographical changes from natural processes (e.g., landslides) or human activities (e.g., mining operations), potentially affecting travel time estimates in dynamic landscapes. The Overture road network dataset may underestimate ghost roads in remote areas (e.g., Southeast Asia's forested regions⁵³) and temporary roads in urban settings⁵⁴. Integrating more multi-source, high-quality datasets are potential solutions for this critical issue. For instance, combining Overture data with local navigation platforms (e.g., Google Maps or Baidu) could improve detection of unlabelled roads in marginalized areas.”

2) The issue of modeling travel time has been addressed by many research teams around the world. Either detailed measurements were carried out and conclusions were formed based on them, or they relied on a variety of assumptions about the speed of movement, direction, etc., to get around under different conditions. The text treats coverage as well as terrain quite generally. This article also does not give enough information about the reliability in determining the class (types) of roads in the countries analyzed. In addition, the assumptions, although based on research and data, are a simplification of reality and may not reflect individual transportation behavior and preferences.

> Thank you for these great comments regarding the methodology. We adopted the friction surface-based approach (Weiss et al. 2018, 2020) to map travel time as a proxy of infrastructure accessibility. This approach assumes the movement speed of humans on Earth's landscape is determined by road infrastructure, water body, topography, and land cover, and is rigorously evaluated and applied for large-scale travel time mapping, such as city accessibility (Weiss and Nelson et al. 2018. Nature) and health facility accessibility (Weiss and Nelson et al. 2020. Nature Medicine). Previous travel time mapping efforts primarily relies on 1-km-resolution friction surface (Weiss et al. 2018, 2020), which cannot sufficiently capture heterogeneous surfaces over urban and mountainous areas. To make this approach more feasible for travel time mapping over urban areas, we created a new high-resolution (30-m resolution) friction surface database by using 10-m-resolution WordCover land cover (Zanaga et al. 2022) and 30-m-resolution ALOS DSM data (Tadono et al. 2014). Our new friction surface provides many more spatial details than the original 1-km-resolution version over urban areas (**Supplementary Figs. 23 and 25**) and mountainous regions (**Supplementary Figs. 26**).

Supplementary Fig. 22. Global maps of friction surface in non-motorized transport (walking mode) that was created by (a) this study, and (b) Weiss's approach. Friction surface in **a** was created from GRIP road network, 10-m-resolution WorldCover land cover, and 30-m-resolution ALOS DSM datasets and resampled to 1-km resolution for visualization. Friction surface in **b** was created from OSM and Google Road networks, 500-m-resolution MCD12Q1 land cover, and 30-m-resolution GMTED2010 DSM datasets with a final 1-km data resolution.

Supplementary Fig. 23 redacted due to lack of 3rd Party Rights

Supplementary Fig. 23. Local examples of friction surface for non-motorized (walking) transport generated by (b and e) this study and (c and f) Weiss's approach over (a-c) Chicago, United States and (d-f) Shanghai, China. a and d, Google satellite image; b and e, friction surface of this study; c and f, friction surface of Weiss's approach.

Supplementary Fig. 24. Global maps of friction surface for the motorized transport that was created by (a) this study, and (b) Weiss's approach. a. Friction surface that was created from GRIP road network, 10-m-resolution WorldCover land cover, and 30-m-resolution ALOS DSM datasets and resampled to 1-km resolution for visualization. **b.** Friction surface that was created from OSM and Google Road networks, 500-m-resolution MCD12Q1 land cover, and 30-m-resolution GMTED2010 DSM datasets with a final 1-km data resolution.

Supplementary Fig. 25 redacted due to lack of 3rd Party Rights

Supplementary Fig. 25. Local examples of friction surface for motorized transport generated by (b and e) this study and (c and f) Weiss's approach over (a-c) Chicago, United States and (d-f) Shanghai, China. a and d, Google satellite image; b and e, friction surface of this study; c and f, friction surface of Weiss's approach.

Supplementary Fig. 26 redacted due to lack of 3rd Party Rights

Supplementary Fig. 26. Local examples of friction surface over two mountainous areas. a-c. Tibetan plateau Mountain. d-f. Andres Mountain. a and d. Google satellite image. b and e. Friction surface map created in this study. c and f. Weiss's friction surface map.

We updated our road network data from GRIP to Overture road data, which provides a total of 12 road

types (i.e., motorway, trunk, railroad, primary, secondary, tertiary, unclassified, residential, service, track, pedestrian, and others; **Supplementary Fig. 19**) with country-level empirical motorized speed limits (**Fig. R2**). Based on Overture road data, we created global friction surface maps for both motorized and non-motorized (i.e., walking) transportation modes using road-specific driving speeds (**Supplementary Fig.20**) and a fixed walking speed of 5 km/h (a common walking speed used by many commercial navigation platforms), respectively. By using Google Directions API, we validated the accuracy of travel time based on our friction surface data over 100 global cities, which shows reliable results for both non-motorized and motorized modes (**Supplementary Figs.27-28**).

Supplementary Fig. 19. Spatial maps of global road lengths of overture road network by class. a. Motorway. b. Trunk. c. Railroad. d. Primary. e. Secondary. f. Tertiary. g. Unclassified. h. Residential. i. Service. j. Track. k. Pedestrian. l. Other.

Fig. R2. Statistical of global road lengths of overture road network by class

Supplementary Fig. 20. Spatial maps of global road speed by class. a. Motorway. b. Trunk. c. Railroad. d. Primary. e. Secondary. f. Tertiary. g. Unclassified. h. Residential. i. Service. j. Track. k. Pedestrian. l. Other. These road speed limitations are adapted from Weiss et al. 2020.

Supplementary Fig. 27 redacted due to lack of 3rd Party Rights

Supplementary Fig. 27. Global sample cities for travel time validation. **a.** Spatial distribution of 100 sample cities (red points). **b-g.** Local examples of six sample cities with 10 validation points (red pentagrams) for each, which are bounded by google urban boundary (GUB) dataset and overlaid on Google satellite basemaps.

Supplementary Fig. 28. Accuracy validation of travel time for (a) non-motorized and (b)

motorized modes. Google travel time is calculated from Google Directions API and our travel time is calculated from the friction surface with the “least-cost-path” algorithm.

This study specifically examines potential physical accessibility (operationalized as minimum travel duration) rather than realized accessibility that incorporates individual mobility behaviors. While factors like transportation mode preferences, route choices, and mobility constraints significantly affect actual travel experiences, such analysis requires: (1) individual-level trajectory data, (2) detailed mobility profiles, and (3) behavioral models - all of which represent important but separate research questions beyond our current infrastructure-focused assessment. Our friction-surface approach provides a critical baseline measure that future studies can build upon by integrating human mobility components.

We have enhanced the descriptions of our methodology regarding road network upgrade (Pages 14-15, Lines 466-486), friction surface map generation (Pages 16, Lines 533-538), and travel time validation in the method section (Pages 16, Lines 551-561) Also discussed the impacts of individual transportation behaviors on actual travel time calculation in the revised manuscript (Page 8, Lines 331-344), which are duplicated as follows:

(Pages 14-15, Lines 466-486) *“Road networks play a crucial role in friction surface mapping, as it largely determines human’s speeds of movements across Earth surface. The Overture road network, primarily built on OpenStreetMap data, provides the most comprehensive global road coverage, with hierarchical classifications with 12 detailed sub-types (i.e., motorway, trunk, railroad, primary, secondary, tertiary, unclassified, residential, service, track, pedestrian, and others; **Supplementary Fig 19**). We converted these individual road vectors into raster layers to characterize the friction surface. Given the absence of a global road width database and variations in worldwide standards, we applied a uniform empirical width of 10 meters, aligning with the spatial resolution of WorldCover land cover, by buffering the Overture road vectors before rasterization. We modelled two baseline transportation modes: non-motorized (walking) and motorized transports, with actual travel behaviour being a combination of both. Speed limits are critical for travel cost calculation. For motorized mode, we assigned country- and road-type-specific speed limits based on OpenStreetMap data, following a previous study⁹. Where country-specific data were unavailable, default global speeds were applied: motorways (105 km/h), trunks (80 km/h), railroads (30 km/h), primary (65 km/h), secondary (60 km/h), tertiary (55 km/h), unclassified (55 km/h), residential (40 km/h), service (30 km/h), tracks (40 km/h), pedestrian (25 km/h), and other roads (45 km/h) (**Supplementary Fig. 20**). For non-motorized mode, a constant walking speed of 5 km/h was adopted, consistent with the prior study⁹. Based on Overture road network, we generated a 1-km-resolution friction surface and compared it with Weiss’s approach⁹. Our results show a significant lower friction values in human settlement areas, supporting that Overture provides denser road coverage (**Supplementary Fig. 21**).”*

(Pages 16, Lines 533-538) *“To demonstrate the necessity of high-resolution friction mapping for local-scale analysis, we compared our results with Weiss et al.’s 1-km-resolution dataset (based on Open Street Map and Google roads, 500-m-resolution MCD12Q1 land cover, and 30-m-resolution Global Multi-resolution Terrain Elevation Data) (**Supplementary Figs. 22-26**). Our datasets capture much finer spatial details in both urban areas (**Supplementary Figs. 23 and 25**) and mountainous regions (**Supplementary Fig. 26**).”*

(Pages 16, Lines 551-561) “To validate the travel time mapping algorithm, we conducted a comparative assessment using the Google Directions API as a reference (<https://developers.google.com/maps/documentation/directions>). First, we randomly selected 100 cities worldwide from the Global Urban Boundary (GUB) dataset³⁵ and sampled 10 locations per city (**Supplementary Fig. 27**). For each city, we then constructed 45 origin-destination (O-D) pairs (i.e., $10 \times 9/2 = 45$), assuming the routes are reversible. Next, we computed travel times for all O-D pairs using both friction surface-based algorithm and the Google Directions API. Comparison results show strong consistency between the two methods for both non-motorized (walking) and motorized travel modes (with $R^2 = 0.94$ for non-motorized mode and $R^2 = 0.79$ for motorized mode; **Supplementary Figs. 28**), confirming the reliability of the proposed friction-based approach.”

(Page 8, Lines 331-344) “Second, while travel time provides a useful proxy for potential physical accessibility (measured as minimum travel duration), actual travel behaviours are constrained by multiple factors, including individual transportation preferences (e.g., cost sensitivity and trip purposes), service quality (e.g., operational capacity and hours), and mobility constraints (e.g., weather conditions and traffic congestion). Our least-cost-path algorithm, which identifies the nearest resources for travel time estimation, assumes people prefer proximate services. However, this assumption may underestimate true accessibility, as actual utilization of daily necessities and services depends on the built environment, socioeconomic status, and personal preferences. Emerging human mobility data sources, such as mobile phone records (e.g., call detail records, Google Maps, and Strava) and social media check-ins (e.g., Twitter and Foursquare), offer valuable individual-level insights into visitation patterns and service usage. Integrating these datasets with spatiotemporal interaction models that account for personal transportation behaviours and preferences could substantially improve our understanding of human access to essential services.”

References:

- Tadono, T., Ishida, H., Oda, F., Naito, S., Minakawa, K., & Iwamoto, H. (2014). Precise global DEM generation by ALOS PRISM. *ISPRS Annals of the Photogrammetry, Remote Sensing and Spatial Information Sciences*, 2, 71-76.
- Weiss, D. J., Nelson, A., Vargas-Ruiz, C. A., Gligorić, K., Bavadekar, S., Gabilovich, E., ... & Gething, P. W. (2020). Global maps of travel time to healthcare facilities. *Nature medicine*, 26(12), 1835-1838.
- Zanaga, D., Van De Kerchove, R., Daems, D., De Keersmaecker, W., Brockmann, C., Kirches, G., ... & Arino, O. (2022). ESA WorldCover 10 m 2021 v200.

Travel duration is used as an indicator of accessibility, which is, of course, a common practice (by the way, I have conducted this kind of research myself). However, accessibility is a more complex concept and can depend on many other factors (cost, quality of service, opening hours, individual preferences, climatic conditions, time of year, etc.). Moreover, a short commute/arrival time alone does not guarantee full access. I also don't know how much consideration was given to the detailed directions of the road network, human travel behavior and differences in demographic structure? Is the “travel time” taken into account getting to the destination and returning home? Or just getting to the destination of the trip? Perhaps shuttle migrations should be discussed on a larger scale? The authors in the presented research did not pay attention to a very important feature of space - its resistance.

> Thank you for your insightful comments on the multifaceted nature of accessibility and the nuances of modeling approaches. We clarify that this study focuses on physical accessibility mapping, where travel duration (measured by time) serves as a proxy based on physical movement resistances (e.g., road infrastructure, topography, and land cover) and resource allocations (e.g., POI distribution). This simplified metric was chosen to enable efficient global-scale assessment of human access to services and infrastructure, balancing feasibility with analytical scope.

We fully agree that actual travel behaviors are influenced by socioeconomic factors (cost, service quality, opening hours, road direction, and individual preferences, etc.), as highlighted by the reviewer. However, these factors are more suited to individual-level or community-scale analyses—a direction beyond this study’s focus but critically important for future work.

To address these gaps, we have expanded the discussion in the revised manuscript (Page 8, Lines 331-344) to explicitly acknowledge: 1) the limitations of travel-time-only metrics and the need for multidimensional accessibility frameworks; and 2) the potential of individual mobility data (e.g., GPS trajectories, travel surveys) to capture behavioral and demographic variability in future studies, which are duplicated as follows:

(Page 8, Lines 331-344) *“Second, while travel time provides a useful proxy for potential physical accessibility (measured as minimum travel duration), actual travel behaviours are constrained by multiple factors, including individual transportation preferences (e.g., cost sensitivity and trip purposes), service quality (e.g., operational capacity and hours), and mobility constraints (e.g., weather conditions and traffic congestion). Our least-cost-path algorithm, which identifies the nearest resources for travel time estimation, assumes people prefer proximate services. However, this assumption may underestimate true accessibility, as actual utilization of daily necessities and services depends on the built environment, socioeconomic status, and personal preferences. Emerging human mobility data sources, such as mobile phone records (e.g., call detail records, Google Maps, and Strava) and social media check-ins (e.g., Twitter and Foursquare), offer valuable individual-level insights into visitation patterns and service usage. Integrating these datasets with spatiotemporal interaction models that account for personal transportation behaviours and preferences could substantially improve our understanding of human access to essential services.”*

Despite my interest in the presented research, I am strongly disturbed in a secondary analysis of it by regional as well as national (and even global) averages. Arguably, they mask significant “local differences” and distinctive contexts. So, in practice, this may mean that the overall conclusions may not be fully “consistent” for a specific community (local/regional) or selected geographic area. I want to emphasize that I am not referring to false conclusions in the sense of violating the logic of the study, but rather to results that are subject to a certain margin of uncertainty and may require further research and verification in these specific contexts.

Knowing the conditions (including the results of research) of Europe or the United States, I am not able to accept the presented results (information) on individual maps. Their graphic quality is insufficient (analogously, descriptive analyses need improvement).

In the text, as well, there are many entries that are either a statement of the authors or have some source (without indicating it), for example: Globally, only 38.7% (2.6 31 billion people) and 50.7 % (3.4 billion people) of the population resides within a 15-minute and 32 30-minute walking distance of

essential daily necessities and services, respectively. These results 33 highlight the urgent need to optimize strategies for planning, allocation, and management of 34 critical infrastructure to promote inclusive and sustainable development.

There are many uncertainties in this passage and statements that can be easily disproved by analyzing the sub-regional or regional scale in each country.

> Thank you for your thoughtful concerns regarding the potential limitations of regional, national, and global averages in masking local variability. To clarify, our country- and state-level analysis aims to address three central questions: (1) How much travel time is required for urban residents to access essential daily necessities and services? (2) What are the global patterns of the "x-minute city" model, and how far we are from achieving it? (3) To what extent does disparity in the travel time to access daily necessities and services vary across geographic, urbanization, and socio-economic spectrums? To address this global question, we adopted a bottom-up approach, first generating fine-resolution friction surface data and calculating pixel-level (30-m resolution) travel times to essential services. These high-resolution estimates were then aggregated across administrative units, from state to country levels. Importantly, our high-resolution travel time dataset also supports local-scale analyses, yielding results consistent with those at regional and global scales. To validate the accuracy of our travel time algorithm at the local level, we conducted a comparative assessment using the Google Directions API (<https://developers.google.com/maps/documentation/directions>) across 100 randomly selected global cities, with 45 origin-destination (O-D) pairs per city. Our validation confirms the high accuracy of our travel time mapping approach (**Supplementary Figs. 27-28**).

Supplementary Fig. 27 redacted due to lack of 3rd Party Rights

Supplementary Fig. 27. Global sample cities for travel time validation. **a**. Spatial distribution of 100 sample cities (red points). **b-g**. Local examples of six sample cities with 10 validation points (red pentagrams) for each, which are bounded by google urban boundary (GUB) dataset and overlaid on Google satellite basemaps.

Supplementary Fig. 28. Accuracy validation of travel time for (a) non-motorized and (b) motorized modes. Google travel time is calculated from Google Directions API and our travel time

is calculated from the friction surface with the “least-cost-path” algorithm.

We acknowledge that country- and state-level analysis may obscure local variations. We selected four local cities in United States (Chicago and Washington; **Supplementary Figs. 8-9** and **12-13**) and Europe (Berlin and Paris; **Supplementary Figs. 10-11** and **14-15**) to demonstrate the spatial details of travel time mapping using our approach that is based on a high-resolution friction surface data. These visualizations highlight the critical influence of spatial configurations, including POI distribution, road networks, and land cover, on accessibility. Notably, areas with denser POIs and road networks exhibit shorter travel times.

Supplementary Fig. 8. Travel time of Chicago by non-motorized mode for six essential daily infrastructure and services. a. Living. b. Healthcare. c. Education. d. Entertainment. e. Public transit. f. Working. Places of interest (POIs) are overlaid on travel time map with purple colors.

Supplementary Fig. 9. Travel time of Washington by non-motorized mode for six essential daily infrastructure and services. a. Living. b. Healthcare. c. Education. d. Entertainment. e. Public transit. f. Working. Places of interest (POIs) are overlaid on travel time map with purple colors.

Supplementary Fig. 10. Travel time of Berlin by non-motorized mode for six essential daily infrastructure and services. a. Living. b. Healthcare. c. Education. d. Entertainment. e. Public transit. f. Working. Places of interest (POIs) are overlaid on travel time map with purple colors.

Supplementary Fig. 11. Travel time of Paris by non-motorized mode for six essential daily infrastructure and services. a. Living. b. Healthcare. c. Education. d. Entertainment. e. Public transit. f. Working. Places of interest (POIs) are overlaid on travel time map with purple colors.

Supplementary Fig. 12. Travel time of Chicago by motorized mode for six essential daily infrastructure and services. a. Living. b. Healthcare. c. Education. d. Entertainment. e. Public transit. f. Working. Places of interest (POIs) are overlaid on travel time map with purple colors.

Supplementary Fig. 13. Travel time of Washington by motorized mode for six essential daily infrastructure and services. a. Living. b. Healthcare. c. Education. d. Entertainment. e. Public transit. f. Working. Places of interest (POIs) are overlaid on travel time map with purple colors.

Supplementary Fig. 14. Travel time of Berlin by motorized mode for six essential daily infrastructure and services. a. Living. b. Healthcare. c. Education. d. Entertainment. e. Public transit. f. Working. Places of interest (POIs) are overlaid on travel time map with purple colors.

Supplementary Fig. 15. Travel time of Paris by motorized mode for six essential daily infrastructure and services. a. Living. b. Healthcare. c. Education. d. Entertainment. e. Public transit. f. Working. Places of interest (POIs) are overlaid on travel time map with purple colors.

We have strengthened the presentation of our results throughout the manuscript by incorporating additional local-level findings and ensuring proper referencing of figures and sources. The sentences noted by the reviewer were originally part of our abstract, which we have now refined them as follows:

“Globally, only 62.8% (3.08 billion people) and 82.5 % (4.04 billion people) of the urban population resides within a 15-minute and 30-minute walking distance of essential daily necessities and services, respectively. These results highlight the urgent need to optimize strategies for planning, allocation, and management of critical infrastructure to promote inclusive and sustainable development”

To make these points clear, we enhanced our local-level analysis, including accuracy validation of travel time mapping (Pages 16, Lines 551-561) and adding detailed case studies of specific cities (Pages 5, Lines 194-197), which are duplicated as follows:

(Pages 16, Lines 551-561) *“To validate the travel time mapping algorithm, we conducted a comparative assessment using the Google Directions API as a reference (<https://developers.google.com/maps/documentation/directions>). First, we randomly selected 100 cities worldwide from the Global Urban Boundary (GUB) dataset³⁵ and sampled 10 locations per city (**Supplementary Fig. 27**). For each city, we then constructed 45 origin-destination (O-D) pairs (i.e., $10 \times 9 / 2 = 45$), assuming the routes are reversible. Next, we computed travel times for all O-D pairs using both friction surface-based algorithm and the Google Directions API. Comparison results show strong consistency between the two methods for both non-motorized (walking) and motorized travel modes (with $R^2 = 0.94$ for non-motorized mode and $R^2 = 0.79$ for motorized mode; **Supplementary Figs. 28**), confirming the reliability of the proposed friction-based approach.”*

(Pages 5, Lines 194-197) *“Local examples across four cities (Chicago, Washington, Berlin, and Paris) show large heterogeneous in travel time maps that are shaped by spatial configurations of infrastructure amenities and transportation networks (Supplementary Figs.8-15).”*

The literature review is not satisfactory, among other things, in terms of analyzing studies on cities of 15, 30 min, etc. A number of concepts and problems have not been satisfactorily explained, e.g. social justice, equitable access, etc. I do not develop this thread because it is the least of the problems of the reviewed text.

There are entries in the study, for me, at a very high level of generality ... almost a popular scientific level. As an example:

Travel time contrast across the urban-rural continuum

> Thank you for these great comments. Following your and Reviewer#1’s suggestions, we enhanced the background and literature reviews of “x-minute” city concepts by involving more related studies on cities of x minutes (Page 3, Lines 76-85). We also improved our explanations of social justice and equitable access to ensure these concepts are correct and clear (Page 2, Lines 41-45). We revised our sentences to increase the academic rigor, and the example sentence was updated as “urban-rural travel time contrasts”.

We have involved these points in the revised manuscript (Page 3, Lines 76-85), which are duplicated as follows:

(Page 3, Lines 76-85) *“A central focus of the “x-minute city” concept is developing holistic methods or metrics to evaluate accessibility to essential services, considering factors such as service availability, road networks, active mobility options, and socioeconomic diversity²³⁻²⁴. Location-based approaches—such as proximity-based measures and the two-step floating catchment area (2SFCA) method—are widely used to assess accessibility, quantifying service availability within specific travel distances or times²⁵. More advanced models, such as spatial-temporal interaction frameworks, incorporate additional dimensions like service capability, attractiveness, user preferences, and human mobility patterns to better reflect actual usage of amenities¹⁶. Among these metrics, proximity-based measures (e.g., accessibility or proximity) remain the most common due to their simplicity and efficiency.”*

(Page 2, Lines 41-45) *“Equitable access to those daily necessities and services (i.e. **individuals have equal opportunity to reach amenities, regardless of socioeconomic disparities in need**) is crucial for enhancing life satisfaction, happiness, and community prosperity, aligning with the United Nations' Sustainable Development Goals (SDGs)³.”*

We examined the difference in travel time between urban and rural areas, which represent the contrast in economic and environmental contexts and is significantly influenced by the geographical distribution of population density, transportation networks, and essential infrastructures (Fig. 4 and Extended Data Fig. 6). Our analysis reveals that among 276 countries, 117 (42.4%) have travel time to daily necessities and services within 15 minutes, 45 186 (16.3%) between 15-30 minutes, 20 (7.2%) between 30-60 minutes, and 7 (2.5%) over 60 minutes respectively, by walking in urban central areas

(Fig. 4a). In comparison, suburban 188 regions experience longer travel time to access essential necessities and services, with 38 189 (13.8%), 57 (20.6%), 36 (13.0%), and 79 (28.6%) countries falling within the 15-minute, 15-30 190 minute, 30-60 minute, and over 60-minute travel time brackets, respectively, by walking (Fig. 191 4b). Conversely, in rural areas, only 72 (26.1%) can access daily necessities and necessities 192 within an acceptable travel time of <60 minutes, while the remaining 150 (54.4%) areas require at least 60 minutes of walking (Fig. 4c). Additionally, we found that disparities in travel time between urban centers and their surrounding areas are more pronounced in regions with lower economic development, regardless of transport modes (Fig. 4d and Extended Data Fig. 6d). For example, the average travel time difference between urban and peri-urban areas is only 19.4 minutes for high-income countries, compared to 196.7 minutes in low-income countries. Likewise, these contrasts in travel time between urban and rural areas are intensified when moving from high-income to low-income countries.

The text quoted above itself, as well as its graphic supplement, on the one hand, does not surprise - it was to be expected. On the other hand, it presents an average value judgement that does not reflect (capture) regional conditions or relationships of interest to, among others, geographer, regionalist, urban planner, etc.

> Thank you for your insightful comments. This study aims to address three core research questions: (1) How much travel time is required for urban residents to access essential daily necessities and services? (2) What are the global patterns of the “x-minute city” model, and how far we are from achieving it? (3) To what extent does disparity in the travel time to access daily necessities and services vary across geographic, urbanization, and socio-economic spectrums? To answer these questions, we developed a novel global framework integrating high-resolution friction surface data, comprehensive POI inventories, and scalable travel-time mapping algorithms. This approach provides the first quantitative, globally comparable assessment of accessibility inequalities—a perspective unattainable through localized studies alone.

These global results do not deny the potential analysis for regional areas that might be interested by local related stakeholders but provides extra holistic and boarder perspective for us to measure and assess the important “x-minute” concept to improve our urban planning and practices. As demonstrated in four detailed city case studies (i.e., Berlin, Paris, Chicago, and Washington; See **Supplementary Figs. 8-15**), the methodology captures both global patterns and local accessibility dynamics at fine spatial resolution. These local-scale analyses deliver actionable insights for municipal governments, urban planners, and infrastructure designers, enabling evidence-based strategies for more equitable and efficient service provision.

To make these points clear, we discussed the potential of this scalable travel time mapping approach to engage local-level policymaking regarding closing urban-rural contrasts in the revised manuscript (Page 7, Lines 272-291), which are duplicated as follows:

“For example, this travel time information can be translated into actionable guidance to address urban-rural contrasts through two key steps: 1) identifying hotspot areas by overlaying travel-time maps with urban-rural population data (readily available to local governments with demographic details) to pinpoint areas where high travel burdens disproportionately affect specific groups; and 2) developing targeted strategies for these hotspot areas through refining infrastructure types (e.g., enhancing necessary infrastructures or road networks in rural areas), optimizing spatial distributions (e.g.,

relocating or adding facilities based on demographic needs), and adopting tailor-tailed interventions to local demographic structures (e.g., increasing elderly-friendly services in aging communities)."

One last comment on the title: Measuring global human access to essential daily necessities and services. From a practical standpoint, everything is clear to the reader, but... Essential daily necessities and services are inherently local, and the title speaks of global access. The study is not concerned with measuring the existence or quality of local services themselves uniformly around the world. Instead, it focuses on measuring the spatial and temporal accessibility of these local services to people at the global level. In other words, it accounts for (measures) how easily (in terms of travel time) people in different parts of the world can reach their local resources and services needed to function.

> We appreciate the reviewer's insightful comment regarding the distinction between local services and global accessibility. To better reflect the study's focus on measuring accessibility (rather than the services themselves), we propose revising the title as *"Measuring global human accessibility to essential daily necessities and services"*.

In conclusion, the reviewed study provides a nice basis for conducting more extensive and effective research in the proposed subject area. It is obvious that one can use in a situation of data scarcity their interpolation - but not in this case.

If you don't have complete data you don't have good research... we all eventually confront this problem in our professional lives.

Unfortunately, I do not recommend this text for publication. Currently, I do not see how it can be revised. I recommend limiting the research space to areas docked with complete and reliable information, and use data in addition: Light Detection and Ranging and/or Digital Elevation Model.

> We greatly appreciate the reviewer's positive comments on the great basis that provided by this study to the field. As we explained above, we adopted a novel scalable framework to map travel time to essential necessities and services by integrating high-resolution friction surface (30-m resolution) and global most comprehensive POI inventories. In this revision, we rigorously validate the reliability of our methodology regarding high-resolution friction surface (comparisons with Weiss et al. global friction surface data; **Supplementary Figs. 21-26**), global POI inventory (cross-comparisons between Overture and Foursquare global POI; **Supplementary Fig. 18**), and travel time mapping algorithm (with reference from Google Directions API; **Supplementary Figs. 27-28**). We also verified our findings by conducting sensitivity analysis with different confidence levels of global POI data (**Supplementary Figs. 5-6**) and providing more local details of travel time mapping over four cities (i.e., Berlin, Paris, Chicago, and Washington; **Supplementary Figs. 8-15**).

Furthermore, we have systematically accounted for topographic effects by integrating 30-meter ALOS DSM data as a key parameter in our friction surface calculations. These terrain adjustments are visually demonstrated in our specific case studies (**Supplementary Fig. 26**), which show how elevation variations influence accessibility patterns.

Supplementary Fig. 26 redacted due to lack of 3rd Party Rights

Supplementary Fig. 26. Local examples of friction surface over two mountainous areas. a-c. Tibetan plateau Mountain. **d-f.** Andres Mountain. **a, d.** Google satellite image. **b, e.** Friction surface created in this study. **c, f.** Weiss's friction surface map.